



# Including filter-feeding gelatinous macrozooplankton in a global marine biogeochemical model: model-data comparison and impact on the ocean carbon cycle

Corentin Clerc[1], Laurent Bopp[1], Fabio Benedetti[2], Meike Vogt[2], and Olivier Aumont[3]

[1]LMD / IPSL, Ecole normale supérieure / Université PSL, CNRS, Ecole Polytechnique, Sorbonne Université, Paris, France
[2]Environmental Physics, Institute of Biogeochemistry and Pollutant Dynamics, ETH Zürich, 8092, Zürich, Switzerland.
[3]LOCEAN / IPSL, IRD, CNRS, Sorbonne Université, MNHN, Paris, France

**Correspondence:** Corentin Clerc (corentin.clerc@lmd.ens.fr)

**Abstract.** Filter-feeding gelatinous macrozooplankton (FFGM), namely salps, pyrosomes and doliolids, are increasingly recognized as an essential component of the marine ecosystem. Unlike crustacean zooplankton (e.g., copepods) that feed on preys that are an order of magnitude smaller, filter-feeding allows FFGM to have access to a wider range of organisms, with predator over prey ratios as high as $10^5$:1. In addition, most FFGM produce carcasses and/or fecal pellets that sink 10 times faster than those of copepods. This implies a rapid and efficient export of organic matter to depth. Even if these organisms represent <5% of the overall planktonic biomass, the induced organic matter flux could be substantial. Here we present a first estimate of the influence of FFGM organisms on the export of particulate organic matter to the deep ocean based on the marine biogeochemical model NEMO-PISCES. In this new version of PISCES, two processes characterize FFGM: the preference for small organisms due to filter feeding, and the rapid sinking of carcasses and fecal pellets. To evaluate our modeled FFGM distribution, we compiled FFGM abundance observations into a monthly biomass climatology using a taxon-specific conversion. A model-observation comparison supports the model ability to quantify the global and large-scale patterns of FFGM biomass distribution, but reveals an urgent need to better understand the factors triggering the boom-and-bust FFGM dynamics before we can reproduce the observed spatio-temporal variability of FFGM. FFGM contribute strongly to carbon export at depth (0.4 Pg C yr$^{-1}$ at 1000 m), particularly in low-productivity region (up to 40% of organic carbon export at 1000 m) where they dominate macrozooplankton by a factor of 2. The FFGM-induced export increases in importance with depth, with a simulated transfer efficiency close to one.

## 1 Introduction

Pelagic tunicates, i.e., salps, doliolids, pyrosomes and appendicularians, are free-swimming open ocean gelatinous zooplankton that are increasingly recognized as key-components of marine ecosystems and biogeochemical cycles (Henschke et al., 2016; Luo et al., 2020). All pelagic tunicates, with the exception of appendicularians, are part of the macrozooplankton (> 2 mm), and are filter-feeding organisms. Hereafter they will be referred to as filter-feeding gelatinous macrozooplankton (FFGM). FFGM, which are urochordates, are water-rich free-swimming transparent animals. Therefore, although they are not part of





the same phyla, they have been placed in the same functional group as ctenophores and cnidarians (jellyfish), i.e. gelatinous zooplankton (GZ).

The fragility of all GZ bodies partly explains the rarity of observations (Henschke et al., 2016). Nevertheless, it has been hypothesized that increasing anthropogenic pressures on the global ocean favor gelatinous zooplankton in most regions due to eutrophication, overfishing, or climate change (Richardson et al., 2009; Purcell, 2012). Research effort focusing on GZ has increased dramatically over the last two decades, particularly on cnidarians ("true-jellyfish") that contribute significantly to biological carbon cycling through "jelly-falls" events (i.e. the accumulation of gelatinous zooplankton carcasses in the water

column following a swarming event; Lebrato et al. (2012); Sweetman et al. (2014); Sweetman and Chapman (2015); Luo et al. (2020)). Similarly, recent studies have also focused on pelagic tunicates (namely salps (e.g., Phillips et al., 2009; Henschke et al., 2020, 2021b, a; Lüskow et al., 2020; Ishak et al., 2020; Stone and Steinberg, 2016) , appendicularians (e.g., Berline et al., 2011) and doliolids (e.g., Stenvers et al., 2021)), revealing their importance in carbon cycling and for the ecosystem structure, at least on a regional scale. Yet, despite this growing interest, the importance of FFGM at the global scale remains uncertain.

Pelagic tunicates are capable of swarming, which means that their population can reach a high abundance in a very short time and can therefore represent a significant part and even dominate the zooplankton community during massive proliferation events (Everett et al., 2011; Henschke et al., 2016). Three mechanisms have been hypothesized to trigger these swarms: i) FFGM use a mucus structure to filter feed which gives them access to a wide range of preys, from bacteria to mesozooplankton (Acuña, 2001; Sutherland et al., 2010; Bernard et al., 2012; Ambler et al., 2013; Sutherland and Thompson, 2022). This

feeding strategy might allow them to proliferate in response to the bloom of a wide variety of organisms, in contrast to typical zooplankton with prey-to-predator size ratios ranging from 1:10 to 1:100 (Hansen et al., 1994). ii) FFGM generally have high clearance and growth rates (Alldredge and Madin, 1982; Henschke et al., 2016) that promote rapid proliferation. The densest FFGM swarms can sweep over 200% of their resident water volume per day (Ishak et al., 2020). iii) Some FFGM, such as salps, have life cycles characterized by the alternation between a sexual phase (the blastozoid) and an asexual phase (the oozoid).

During the asexual phase, oozoids produce long chains of blastozooids clones that can number several hundreds of individuals and give rise to swarming processes (Loeb and Santora, 2012; Kelly et al., 2020; Groeneveld et al., 2020). Based on their potential to form large swarms, FFGM can significantly affect ecological processes, at least locally.

FFGM could also have an impact on the ocean carbon cycle. Indeed, many FFGM produce fast sinking carcasses and/or fecal pellets that induce a very efficient carbon export, especially during swarming events (Henschke et al., 2016). Large fecal pellets

and carcasses of salps are carbon-rich (more than 30% of dry weight (DW)) and sink at speeds up to 2700 m d$^{-1}$ for fecal pellets and 1700 m d$^{-1}$ for carcasses (Henschke et al., 2016; Lebrato et al., 2013). In areas where salps proliferate, they can induce a carbon transfer to the seafloor 10 times faster than in their absence (Henschke et al., 2016). For pyrosomes, knowledge on their impact and the nature of their carcasses and fecal pellets remains very limited (Décima et al., 2019). Intense carcass fall events have been described as responsible for large carbon exports due to their high carbon content (35% DW, one of the

highest among GZ) (Lebrato and Jones, 2009). Although their fecal pellets sink 30 times slower than those of large salps (70 m d$^{-1}$ according to Drits et al. (1992)), they are able to export a significant amount of carbon in addition to active transport through diurnal vertical migrations (Stenvers et al., 2021; Henschke et al., 2019). Because of their rapidly sinking fecal pellets



(over 400 m d$^{-1}$) and high clearance rates, doliolids also affect carbon fluxes (Takahashi et al., 2013, 2015; Ishak et al., 2020) but their impact remains poorly documented.

Overall, most studies to date have focused on regional scales. Recently Luo et al. (2020) have estimated the contribution to the global carbon cycle of three categories of gelatinous zooplankton: ctenophores, cnidarians and pelagic tunicates. Using a data-driven carbon cycle model, they found that pelagic tunicates contribute three quarters of the particulate organic carbon (POC) flux induced by gelatinous zooplankton or one quarter of the total POC exported at 100 m. A more recent study by the same team (Luo et al., 2022) revised this estimate to 0.57 Pg C yr$^{-1}$, representing 9% of total export at 100 m, by explicitly

representing FFGM in the COBALT-v2 biogeochemical model (FFGM refer to large pelagic tunicates in their study).

   Marine biogeochemical models have repeatedly shown their usefulness in understanding marine processes on a global scale: in particular for the role of plankton in ecosystem processes (e.g., Sailley et al., 2013; Le Quéré et al., 2016; Kearney et al., 2021) and biogeochemical fluxes (e.g., Buitenhuis et al., 2006; Kwiatkowski et al., 2018; Aumont et al., 2018). Their complexity has been increased by the addition of multiple limiting nutrients and multiple functional groups or size classes of

phytoplankton and zooplankton (e.g., Le Quéré et al., 2005; Follows et al., 2007; Ward et al., 2012; Aumont et al., 2015). Notably, Plankton Functional Type (PFT) models have been introduced as a way of grouping organisms that keeps the overall biological complexity at a manageable level (Moore et al., 2001; Gregg et al., 2003; Le Quéré et al., 2005). Wright et al. (2021) showed that the introduction of a jellyfish PFT (cnidarians only) into the PLANKTOM model has a large direct influence on the biomass distribution of the crustacean macrozooplankton PFT and indirectly influences the biomass distributions of

protozooplankton and mesozooplankton through a trophic cascade. This influence could be explained by the specific diet of jellyfish that differs from other zooplankton PFTs. Similarly, the inclusion of FFGM as a new PFT in a PFT-based model has been recently achieved by Luo et al. (2022). In their study, they introduced two tunicates groups into the COBALT-v2 model: a large salps/doliolids (FFGM) and a small appendicularian, and they estimated their impact on the surface carbon cycle, but they did not consider the impacts on the deeper carbon cycle.

Here, we use the PISCES-v2 model (Aumont et al., 2015) which is the standard marine biogeochemistry component of NEMO (Nucleus for European Modelling of the Ocean). In this study, a new version of PISCES was developed (PISCES-FFGM) in which two new PFTs were added: a generic macrozooplankton (GM) based on an allometric scaling of the existing mesozooplankton and a filter-feeding gelatinous macrozooplankton (FFGM). Two processes characterize FFGM in this version of the model: access to a wide range of preys through filter feeding and rapid sinking of carcasses and fecal pellets. We

first examine how the model succeeds in reproducing the surface distribution of FFGM by providing a new compilation of abundance observations converted to carbon biomass via taxon-specific conversion functions to make this assessment. Second, because the modeling study by Luo et al. (2022) focused on the impact of FFGM on surface processes, we investigated whether our modeling framework and formulations produce results consistent with theirs. Our study provides also some new insights: 1) we explore the FFGM-specific spatial patterns of organic matter production, export and particles composition in the top

100 m; 2) we investigate the impacts of FFGM on the export of particulate organic carbon to the deep ocean via an explicit representation of fast-sinking fecal pellets and carcasses.



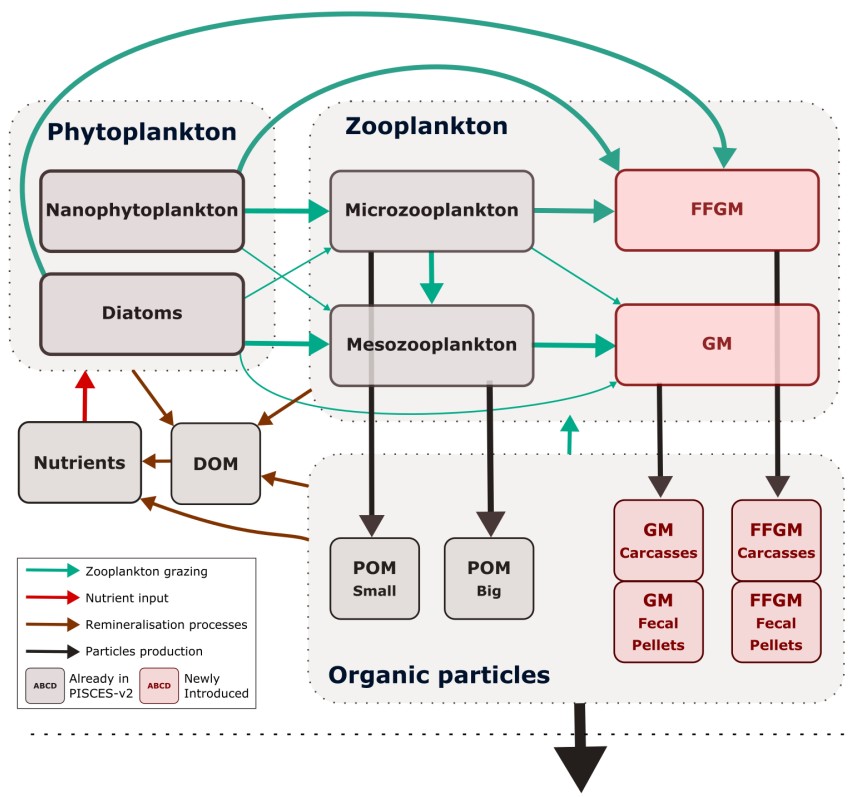

**Figure 1. Architecture of PISCES-FFGM.** This figure only shows the organic components of the model omitting thus oxygen and the carbonate system. This diagram emphasizes trophic interactions (turquoise arrows, the width representing the preference of the predator for the prey) as well as particulate organic matter production (black arrows), two processes strongly impacted by the introduction of two new zooplankton groups in PISCES-FFGM (pink boxes). FFGM is for Filter-Feeding Gelatinous Macrozooplankton, GM is for Generic Macrozooplankton, POM is for Particulate Organic Matter, DOM is for Dissolved Organic Matter.

## 2 Materials and method

### 2.1 Model description

#### 2.1.1 Model structure

The marine biogeochemical model used in the present study is a revised version of PISCES-v2 (gray boxes in fig. 1). It includes five nutrient pools ($Fe$, $NH_4^+$, $Si$, $PO_4^{3-}$ and $NO_3^-$), two phytoplankton groups (Diatoms and Nanophytoplankton, denoted $D$ and $N$), two zooplankton size classes (Micro- and Mesozooplankton, denoted $Z$ and $M$) and an explicit representation of particulate and dissolved organic matter, reaching a total of 24 prognostic variables (tracers). A full description of the model is provided in Aumont et al. (2015).





In the version used here, two groups of macrozooplankton were added, one corresponding to generic macrozooplankton organisms (hereafter referred to as GM, see fig. 1) and the other to salp-like filter-feeding gelatinous macrozooplankton organisms (hereafter referred to as FFGM, see fig. 1). As with micro- and mesozooplankton in the standard version of PISCES, the C:N:P stoichiometric composition of the two macrozooplankton groups is assumed to be constant. In addition to their carbon biomass, two additional tracers were introduced into the model for each macrozooplankton group corresponding to fecal pel-

lets and carcasses in carbon units, respectively (GM Carcasses, GM Fecal Pellets, FFGM Carcasses and FFGM Fecal Pellets, see fig. 1). Because both macrozooplankton groups have a constant Fe:C stoichiometry and feed on phytoplankton that have a flexible Fe:C stoichiometry (Eq. 16 to 20 in Aumont et al. (2015)), two compartments representing the iron content of the fecal pellets of the two macrozooplankton groups were added. Figure 1 summarizes the tracers and interactions newly introduced into PISCES for this study (referred to as PISCES-FFGM hereafter).

In total, the tracers considered for particulate and dissolved organic matter are (organic particles in fig. 1): $sPOC$ which refers to the carbon content of small organic particles, $bPOC$ which refers to the carbon content of large organic particles, $DOC$ which refers to dissolved organic carbon, $Ca_{FFGM}$ which refers to the carbon content of FFGM carcasses, $Fp_{FFGM}$ which refers to the carbon content of FFGM fecal pellets, $Ca_{GM}$ which refers to the carbon content of GM carcasses and $Fp_{GM}$ which refers to the carbon content of GM fecal pellets.

**2.1.2  Macrozooplankton (FFGM and GM) dynamics**

We first present the generic equation describing the dynamics of the two groups of macrozooplankton, and then focus on the modeling choices differentiating these two groups. All symbols and definitions are summarized in Table 1.

The temporal evolution of the two compartments of macrozooplankton is governed by the following equation:

$$
\begin{aligned}
\frac{\partial X}{\partial t} =\ & e^X G_X \left(1 - \Delta(O_2)\right) f_X(T) X \\
& -(m^X + m_c^X) f_X(T) \left(1 - \Delta(O_2)\right) X^2 \\
& -r^X f_X(T) \left(\frac{X}{K_m + X} + 3\Delta(O_2)\right) X
\end{aligned}
\tag{1}
$$

This equation is similar to the one used for micro- and mesozooplankton in PISCES-v2 (Aumont et al., 2015). In this equation, $X$ is the considered macrozooplankton biomass ($GM$ or $FFGM$), and the three terms on the right-hand side represent growth, quadratic and linear mortalities. $e^X$ is the growth efficiency. It includes a dependence on food quality as presented

in PISCES-v2 (Eq. 27a and 27b in Aumont et al. (2015)). Quadratic mortality is divided into mortality due to predation by unresolved higher trophic levels (with a rate $m^X$) and mortality due to disease (with a rate $m_c^X$). All terms in this equation were given the same temperature sensitivity $f_X(T)$ using a Q10 of 2.14 (Eq. 25a and 25b in Aumont et al. (2015)), as for mesozooplankton in PISCES-v2 and according to Buitenhuis et al. (2006). Growth rate and quadratic mortality are reduced and linear mortality is enhanced at very low oxygen levels, as we assume that macrozooplankton are not able to cope with

anoxic waters ($\Delta(O_2)$ varies between 0 in fully oxic conditions and 1 in fully anoxic conditions, see Eq. 57 in Aumont et al. (2015)).



| Symbol | Description |
| --- | --- |
| | **I. STATE VARIABLES** |
| $P$ | Nanophytoplankton |
| $D$ | Diatoms |
| $Z$ | Microzooplankton |
| $M$ | Mesozooplankton |
| $GM$ | GM |
| $FFGM$ | FFGM |
| $Ca_{FFGM}$ | FFGM Carcasses |
| $Fp_{FFGM}$ | FFGM Fecal Pellets |
| $Ca_{GM}$ | GM Carcasses |
| $Fp_{GM}$ | GM Fecal Pellets |
| | |
| | **II. PHYSICAL VARIABLES** |
| $T$ | Temperature |
| | |
| | **III. GROWTH** |
| $e^X$ | growth efficiency of $X$ |
| $a^X$ | unassimilation rate of $X$ |
| $g_m^X$ | maximal $X$ grazing rate |
| $K_G^X$ | half saturation constant for $X$ grazing |
| $p_Y^X$ | $X$ preference for group $Y$ |
| $Y_{\text{thresh}}^X$ | group $Y$ threshold for $X$ |
| $F_{\text{thresh}}^X$ | feeding threshold for $X$ |
| $w_X$ | sinking velocity of $X$ particles |
| $\text{ff}_m^X$ | $X$ flux feeding rate |
| $m^X$ | $X$ quadratic mortality |
| $m_c^X$ | $X$ non predatory quadratic mortality |
| $r^X$ | $X$ linear mortality |
| $K_m$ | half saturation constant for mortality |
| $\alpha$ | remineralisation rate |

**Table 1.** Variables and parameters used in the set of equations governing the temporal evolution of the state variables

The difference between the two macrozooplankton groups lies in the description of the term $G_X$, i.e. the ingested matter. A full description of the equations describing $G_X$ is provided in the appendix Text A3 (Eq. A1 to Eq. A12). Below, we present




the two different choices of feeding representation that differentiate the dynamics of the two macrozooplankton groups, GM
and FFGM.

GM, namely generic macrozooplankton, is intended to represent crustacean macrozooplankton, such as euphausids or large copepods. The parameterization is similar to that of mesozooplankton (Eq. 28 to 31 in Aumont et al. (2015)). Therefore, in addition to conventional suspension feeding based on a Michaelis-Menten parameterization with no switching and a threshold (Eq. A1, A2 and A3), flux-feeding is also represented (Eq. A4) as has been frequently observed for both meso- and macrozoo-
plankton (Jackson, 1993; Stukel et al., 2019). GM can flux-feed on small and large particles as well as on carcasses and fecal pellets produced by both GM and FFGM (Eq. A6). We assume that the proportion of flux-feeders is proportional to the ratio of potential food available for flux feeding to total available potential food (Eq. A7 and A8). Suspension feeding is supposed to be controlled solely by prey size, which is assumed to be about 1 to 2 orders of magnitude smaller than that of their preda-tors (Fenchel, 1988; Hansen et al., 1994). Thus, GM preferentially feed on mesozooplankton, but also, to a lesser extent on
microzooplankton, large phytoplankton and small particles (Eq. A5 and A10, Fig. 1).

FFGM represent the large pelagic tunicates (i.e. salps, pyrosomes and doliolids but not appendicularians). Pelagic tunicates are all highly efficient filter feeders and thus have access to a wide range of prey sizes, from bacteria to mesozooplankton (Acuña, 2001; Sutherland et al., 2010; Bernard et al., 2012; Ambler et al., 2013). There is no strong evidence that FFGM feed on mesozooplankton in the literature. Therefore, we assume in our model that FFGM are solely suspension feeders (i.e. with
concentration-dependent grazing based on a Michaelis-Menten parameterization with no switching and a threshold, see Eq. A1, A2 and A3) feeding with identical preferences on both phytoplankton groups ($D$ and $N$) as well as on microzooplankton ($Z$) (Eq. A11 and A12, Fig. 1). They can also feed on small particles ($sPOC$, Sutherland et al. (2010)) (Eq. A11, Fig. 1).

### 2.1.3 Carcasses and fecal pellet dynamics:

Carcasses $Ca_{FFGM}$ and $Ca_{GM}$ are produced as a result of non predatory quadratic and linear mortalities of GM and FFGM,
respectively. The $Fp_{FFGM}$ and $Fp_{GM}$ are produced as a fixed fraction of the total food ingested by the two macrozooplankton groups. Remineralization of fecal pellets and carcasses by bacteria is modeled using the same temperature-dependent specific degradation rate with a $Q_{10}$ of 1.9, identical to that used for small and large particles. In addition to remineralization, carcasses and fecal pellets undergo flux feeding by GM as explained in the previous subsection. Note that parasitism is not considered in this study because it is too poorly documented, but that it could represent an important source of carcasses (Lavaniegos and
Ohman, 1998; Phleger et al., 2000; Hereu et al., 2020). The sinking speeds of these particle pools are assumed to be constant. A complete description of the equations governing the temporal evolution of fecal pellets and carcasses is provided in the appendices section Text A3 (Eq. A14 and A15).





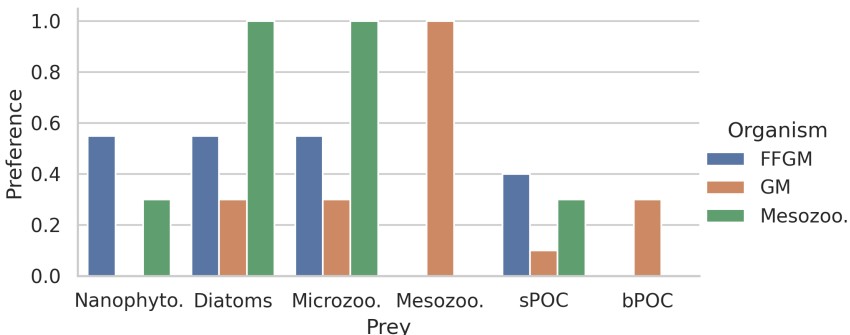

**Figure 2. Histogram of the preferences of secondary consumers for their respective prey.** Secondary consumers are mesozooplankton, FFGM and GM, and preys are nanophytoplankton, diatoms, microzooplankton, mesozooplankton, small organic particles and large organic particles. A preference of 1 indicates that any prey reached is consumed, a preference of 0 indicates that the prey is never consumed.

## 2.2 Standard experiment

### 2.2.1 Model parameters

Each zooplankton group is characterized by a size range, assuming that sizes within the group are distributed along a spectrum of constant slope -3 in log-log space, according to the hypothesis of Sheldon et al. (1972). These ranges are: 10-200 $\mu$m for microzooplankton, 200-2000 $\mu$m for mesozooplankton and 2000-20000 $\mu$m for macrozooplankton (GM and FFGM).

All parameters in PISCES-FFGM have identical values to those in Aumont et al. (2015). The only exception is the mesozooplankton quadratic mortality rate, whose value has been greatly reduced from its standard value of $3e4\,L\,mol^{-1}\,d^{-1}$ to

$4e3\,L\,mol^{-1}\,d^{-1}$ since predation by higher trophic levels is now explicitly represented.

The parameter values that were introduced in PISCES-FFGM to represent the evolution of GM and FFGM are given in Table 2. Metabolic rates are assumed to vary with size according to the allometric relationship proposed by Hansen et al. (1997). Therefore, maximum grazing, respiration and flux-feeding rates were calculated from their values for mesozooplankton using a size ratio of 10. The preferences of GM and FFGM for their different prey are detailed in section 2.1.2. Their values are shown

in Figure 2. The sinking speed of FFGM carcasses (resp. fecal pellets) is set to $800\,m\,d^{-1}$ (resp. $1000\,m\,d^{-1}$) (Henschke et al., 2016). The sinking speeds of GM fecal pellets and carcasses are set rather arbitrarily to $100\,m\,d^{-1}$ and $300\,m\,d^{-1}$ respectively, within the wide range of values found in the literature (Small et al., 1979; Fowler and Knauer, 1986; Lebrato et al., 2013; Turner, 2015). The quadratic mortality rates have been adjusted by successive simulations evaluated against the observations presented in the next section.





| Symbol | Source | GM ($X = GM$) | FFGM ($X = FFGM$) | Unit |
|---|---|---|---|---|
| $e^X_{max}$ | ⋆ | 0.35 | 0.35 | - |
| $a^X$ | ⋆ | 0.3 | 0.3 | - |
| $g^X_m$ | ● | 0.28 | 0.28 | d$^{-1}$ |
| $K^X_G$ | ⋆ | 2e-5 | 2e-5 | mol L$^{-1}$ |
| $p^P_X$ | ‡ | 0 | 0.55 | - |
| $p^D_X$ | ‡ | 0.3 | 0.55 | - |
| $p^Z_X$ | ‡ | 0.3 | 0.55 | - |
| $p^M_X$ | ‡ | 1 | 0 | - |
| $p^{POC}_X$ | ‡ | 0.1 | 0.4 | - |
| $p^{GOC}_X$ | ‡ | 0.3 | 0 | - |
| $P^X_{thresh}$ | ⋆ | 1e-8 | 1e-8 | mol L$^{-1}$ |
| $D^X_{thresh}$ | ⋆ | 1e-8 | 1e-8 | mol L$^{-1}$ |
| $Z^X_{thresh}$ | ⋆ | 1e-8 | 1e-8 | mol L$^{-1}$ |
| $M^X_{thresh}$ | ⋆ | 1e-8 | 1e-8 | mol L$^{-1}$ |
| $POC^X_{thresh}$ | ⋆ | 1e-8 | 1e-8 | mol L$^{-1}$ |
| $F^X_{thresh}$ | ⋆ | 3e-7 | 3e-7 | mol L$^{-1}$ |
| $w_{Ca_X}$ | ‡ | 300 | 800 | m d$^{-1}$ |
| $w_{Fp_X}$ | ‡ | 100 | 1000 | m d$^{-1}$ |
| ff$^H_m$ | ● | 5e5 | - | m$^2$ mol$^{-1}$ |
| $m^X$ | † | 1.2e4 | 1.2e4 | L mol$^{-1}$ d$^{-1}$ |
| $m^X_c$ | † | 4e3 | 4e3 | L mol$^{-1}$ d$^{-1}$ |
| $r^X$ | ● | 0.003 | 0.005 | d$^{-1}$ |
| $K_m$ | ⋆ | 2e-7 | 2e-7 | mol L$^{-1}$ |
| $\alpha$ | ⋆ | 0.025 | 0.025 | d$^{-1}$ |

**Table 2. Parameter values used in PISCES-FFGM.** The symbols in the "Source" column indicate how the parameter value was determined: (⋆) parameters for which we assumed that both GM and FFGM share the same characteristics as mesozooplankton, (●) metabolic rates assumed to vary with size, thus scaled using an allometric scaling convertion of mesozooplankton value based on (Hansen et al., 1997), (†) parameters tuned to fit PISCES-v2 general biology dynamics, and (‡) indicates parameters whose values have been arbitrarily set based on information available in the literature and/or of the authors expertise.

### 2.2.2 Reference simulation

The biogeochemical model is run in an offline mode with dynamical fields identical to those used in Aumont et al. (2015). These climatological dynamic fields (as well as the input files) can be obtained from the NEMO website (www.nemo-ocean.eu) and were produced using an ORCA2-LIM configuration (Madec, 2008). The spatial resolution is about $2°$ by $2° \cos(\phi)$ (where





$\phi$ is the latitude) with a meridional resolution enhanced at $0.5°$ in the equator region. The model has 30 vertical layers with
increased vertical thickness from 10 m at the surface to 500 m at 5000 m. PISCES-FFGM was initialized from the quasi-
steady-state simulation presented in Aumont et al. (2015). The two macrozooplankton groups, their fecal pellets and carcasses
were set to a small uniform value of $10^{-9}$ mol C L $^{-1}$. The model was then integrated for the equivalent of 600 years, forced
with 5-day averaged ocean dynamic fields and with a three-hour integration time step.

## 2.3   Sensitivity experiments

The first experiment, PISCES-GM ("Generic Macrozooplankton"), was designed to investigate the impact of an explicit FFGM
representation (with a different grazing parameterization than GM) on the spatial and vertical distribution of POC fluxes. In
PISCES-GM, the FFGM ingestion rate ($g_m^{FFGM}$ defined in table 1 and used in Eq. A3) was set to 0 which is equivalent to
running the model with a single generic macrozooplankton group.

The second experiment, PISCES-HGR ("High Growth Rate"), was designed to investigate the impact of higher clearance
rates observed for FFGM than for GM. In PISCES-HGR, the FFGM ingestion rate ($g_m^{FFGM}$ defined in table 1 and used in Eq.
A3) was set to 1.4 d$^{-1}$, which corresponds to a high value of the range provided by Luo et al. (2022) (0.105-1.85 d$^{-1}$).

The third experiment, PISCES-HGM ("High Growth and Mortality rates"), is similar to PISCES-HGR, but tries to compen-
sate the unrealistic high biomasses induced by FFGM high clearance rates in PISCES-HGR. To do so, non-predatory ($m_c$)
quadratic mortality was increased so that FFGM biomass on the upper 100 m is similar to PISCES-FFGM. The quadratic
mortality due to predation was not modified because there is no reason to believe that FFGM are subject to a higher predation
pressure than GM.

The fourth experiment, PISCES-LOWV ("Low Velocity"), was designed to evaluate the impact of the high sinking speeds
of particles from GM and FFGM. In PISCES-LOWV, the sinking speeds of all fecal pellets and carcasses produced by GM and
FFGM ($w_{Fpx}$ and $w_{Cax}$, defined in table 1 and used in Eq. A14 and A15) were assigned the same values as for large particles
in PISCES-v2, i.e. 30 m d$^{-1}$.

The fifth experiment, PISCES-CLG ("Clogging"), was designed to explore the impacts of clogging. Clogging, defined as
the saturation of an organism's filtering apparatus with high levels of particulate matter, is a poorly documented mechanism
for FFGM but has been observed (Harbison et al., 1986; Perissinotto and Pakhomov, 1997) or suggested (Perissinotto and
Pakhomov, 1998; Pakhomov, 2004; Kawaguchi et al., 2004) for some salps species. Unlike other macrozooplankton groups,
it has been shown that salps biomass remain relatively low at high chlorophyll concentrations (Heneghan et al., 2020). In
PISCES-CLG, the achieved ingestion rate of FFGM ($G_{FFGM}$, see Eq. A13) is modulated by a clogging function $F_C(Chl)$
inspired by the parameterization proposed by Zeldis et al. (1995):

$$F_C(Chl) = 1 - \frac{1}{2}\left(1 + \mathrm{ERF}(C_{sh}(NCHL + DCHL - C_{th}))\right) \qquad (2)$$

In this equation, $C_{th}$ is the clogging threshold, $C_{sh}$ is the clogging sharpness and ERF is the Gauss error function. A low
clogging threshold $C_{th}$ of 0.5 mg Chl m$^{-3}$ is chosen to limit FFGM growth in all moderate and high productivity regions.
Clogging sharpness $C_{sh}$ is set to 5 mg Chl m$^{-3}$, the value proposed by Zeldis et al. (1995).



Values of the parameters that were changed in the five sensitivity experiments are presented in Table 3. All five sensitivity experiments were initialized with the year 500 output fields from the baseline PISCES-FFGM experiment. They were then run for 100 years. All results presented in this study are average values over the last 20 years of each simulation. PISCES-

CLG, PISCES-HGR and PISCES-HGM help to investigate the modeled distribution of GM and FFGM while PISCES-GM and PISCES-LOWV are used for exploring the spatial pattern and depth gradient of particulate organic carbon fluxes.

| Experiment | PISCES-FFGM (Standard) | PISCES-GM | PISCES-HGR | PISCES-HGM | PISCES-LOWV | PISCES-CLG |
|---|---|---|---|---|---|---|
| FFGM maximal growth rate | 0.28 d$^{-1}$ | 0 d$^{-1}$ | 1.4 d$^{-1}$ | 1.4 d$^{-1}$ | - | - |
| FFGM non-predatory quadratic mortality | 4e3 L mol$^{-1}$d$^{-1}$ | - | - | 1e5 L mol$^{-1}$d$^{-1}$ | - | - |
| Carcasses and Fecal pellets sinking velocities | 100-1000 m d$^{-1}$ | - | - | - | 30 m d$^{-1}$ | - |
| Clogging threshold | $\infty$ | - | - | - | - | 0.5 mg Chl m$^{-3}$ |

**Table 3. Sensitivity experiments parameterization.** A dash indicates that the parameter value is the same as in the standard PISCES-FFGM experiment.

## 2.4 Observations

### 2.4.1 FFGM biomass estimates

We compiled an exhaustive dataset of in situ pelagic tunicates (i.e., Thaliaceans) concentrations from large scale plankton
monitoring programs and previous plankton data compilations to derive monthly field of pelagic tunicates biomass (in mg C m$^{-3}$). This product can be used as a standard data set to evaluate the FFGM biomass estimated by PISCES-FFGM. First, three main data sources were retrieved: NOAA's Coastal and Oceanic Plankton Ecology, Production, and Observation Database (COPEPOD; O'Brien (2014)), the Jellyfish Database Initiative (JeDI; Lucas et al. (2014)), KRILLBASE (Atkinson et al., 2017). The Australian Continuous Plankton Recorder (CPR) survey (AusCPR; IMOS (2021)) and the Southern Ocean CPR survey
(SO-CPR; (Hosie, 2021)) were excluded because they were found to not quantitatively sample thaliaceans (see appendices, Text A1). This compilation gathered planetary scale plankton concentration measurements collected through a broad variety of sampling devices over the last 100 years, with taxonomic identification of varying precision and scientific names, some of which changed through time. Therefore, we curated the scientific names and the taxonomic classification of each observation to harmonize names across all data sets and to correct deprecated names and synonyms based on the backbone classification
of the World Register of Marine Species (WoRMS; Horton et al. (2022)) using the 'worms' R package version 0.2.2 (Holstein, 2018). Then, only those observations corresponding to an organism belonging to the Class Thaliacea were kept. Observations without a precise sampling date and at least one sampling depth indicator (usually maximum sampling depth, in meters) were discarded. All data sets provided concentrations in ind m$^{-3}$ except KRILLBASE that provided salp (mostly *Salpa thompsoni*) densities in ind m$^{-2}$, which we converted to ind m$^{-3}$ based on the maximum sampling depth of the corresponding net tows.
In KRILLBASE, 5'186 observations of Thaliaceans with missing density values were discarded (35.6% of the original 14'543 observations). In COPEPOD, concentrations are standardized as if they were all taken from a plankton net equipped with a





330 $\mu$ m mesh (Moriarty and O'Brien, 2013). 862 point observations with missing concentration values were discarded (3.5%

of the original 24'316 observations). We examined the composition of the original data sources compiled within JeDI and

COPEPOD by assessing the recorded institution codes as well as their corresponding spatio-temporal distributions to evaluate

the observations overlapping between these two previous data syntheses. We logically observed a very high overlap between

COPEPOD and JeDI as the former data set was the main data contributor to the latter. Therefore, overlapping records were

identified based on their sampling metadata, scientific names, concentration values, the recorded institution codes and recorded

data sources, and were removed from JeDI. This removed 14'198 (74.1%) of the JeDI's original Thaliaceans observations.

This synthesis of Thaliaceans concentrations gathered globally distributed 34'566 point observations (Figure A1), collected

at a mean ($\pm$ std) maximum sampling depth of 193 ($\pm$ 198) m over the 1926-2009 time period (mean $\pm$ std of the sampling

year is 1975.9 $\pm$ 19.3). The range of observed Thaliacean concentration ranged from 0.0 ind m$^{-3}$ to 10'900 ind m$^{-3}$ with an

average of 4.2 ($\pm$ 103) ind m$^{-3}$.

Most of the records showed a fairly precise taxonomic resolution as 1.6% of the data was species-resolved (mostly *S.*

*thompsoni*, *Soestia zonaria*, *S. fusiformis* and *Thalia democratica*), 42% genus-resolved (mostly *Thalia*, *Doliolum* and *Salpa*)

and 83% family-resolved (mostly Salpidae and Doliolidae). Therefore, we were able to perform taxon-specific conversions

from individual concentrations to biomass concentrations (in mg C m$^{-3}$) for each point observation (see Table A1). We used

the taxon-specific carbon weights (mg C ind$^{-1}$) summarized by Lucas et al. (2014), which were based on the group-specific

length–mass or mass–mass linear and logistic regression equations of Lucas et al. (2011). Not all the observations had a

precise counter part in the carbon weights compilation of Lucas et al. (2014) because they were not identified at the species

or the genus level (e.g., Class-level, Order-level or Family-level observations). In these cases, we computed the median carbon

weight of those taxa reported in Lucas et al. (2014) and which composed the higher level taxonomic group (i.e., the carbon

weight of Salpidae corresponded to the average carbon weight of all Salpidae species), and used this average carbon weight

to convert the individual concentrations to carbon concentrations. Biomass observations larger than two times the standard

deviation were considered as outliers and were excluded. Then, we only retained observations from the upper 300 m to exclude

deep water samples and focus on zooplankton communities that inhabit the euphotic layer. The biomass levels of this subset

ranged between 0.0 and 488 mg C m$^{-3}$ (4.9 $\pm$ 25.7 mg C m$^{-3}$). Thaliacean concentrations issued from single net sample were

summed when necessary (e.g., when species and/or genera counts were sorted within one plankton sample) to be representative

of a Thaliacea-level point measurement. At this point, the dataset contains 18'875 single observation of Thaliacean biomass.

Hereafter, we will refer to this dataset as "AtlantECO dataset".

Ultimately, monthly Thaliacean biomass fields were computed for validating the monthly FFGM biomass fields of PISCES-

FFGM. Thaliacea biomass concentrations were averaged per months on a 72x36 grid to obtain the 12 monthly climatological

fields of Thaliacea biomass needed for evaluating our model. A low resolution grid (5°x 5°) has been used to counterbalance

patchiness of data, as suggested by Lilley et al. (2011). After this final step, the monthly climatological values of Thaliacea

biomass concentrations ranged between 0.0 and 454 mg C m$^{-3}$ (6.53 $\pm$ 26.21 mg C m$^{-3}$). Hereafter, we will refer to this

climatology as "AtlantECO climatology".



### 2.4.2 Additional datasets

We also used the monthly fields derived from observations as a standard data set to evaluate some of the other PISCES-FFGM compartments: total macrozooplankton, mesozooplankton, total chlorophyll, nutrients and oxygen.

As with FFGM, for total macrozooplankton observations, a low resolution grid has been used. We use monthly macrozoo-
plankton abundances binned on a 72x36 grid (ind m$^{-3}$, vertically integrated between 0 and 100 m) from MARine Ecosystem DATa (MAREDAT) (Moriarty et al., 2013), and then convert abundances to carbon-based concentration to evaluate our modeled distribution of total macrozooplankton biomass (i.e. FFGM and GM). To convert to carbon concentration, an average individual weight of 588 $\mu$g was chosen by considering an individual with a mean size of 6.3 mm (the geometric mean of the macrozooplankton size class) and applying the relationship proposed for copepods by Watkins et al. (2011).

Monthly observations fields were binned on a 360x180 grid to validate other modeled distributions. The mesozooplankton field (mmol m$^{-3}$, vertically integrated between 0 and 300 m) from MARine Ecosystem DATa (MAREDAT) (Moriarty and O'Brien, 2013) is used to evaluate our modeled total mesozooplankton biomass distribution. The $PO_4^{3-}$ and $NO_3^-$ surface fields from the World Ocean Atlas (Garcia et al., 2019) are used to evaluate our modeled nutrient distributions. The long-term multi-sensor time-series OC-CCI (Ocean Colour project of the ESA Climate Change Initiative, Sathyendranath et al. (2019)) of
satellite phytoplankton chlorophyll-$a$ sea surface concentration converted into mg Chl m$^{-3}$ is used to evaluate our modeled total chlorophyll distribution. The same product regridded on a 72x36 grid is used to compare observed and modeled relationships between chlorophyll and FFGM abundance (Fig. 5).

### 2.4.3 Model evaluation

The model evaluation is based on monthly fields averaged over the last 20 years of the PISCES-FFGM reference simulation.
For each unique observation in the AtlantECO dataset, we sampled the modeled FFGM biomass from the PISCES-FFGM climatology at the corresponding coordinates (latitude,longitude), month, and depth range (minimal depth and maximal depth), so that each observed biomass can be compared to a "model-sampled" biomass. When compared to the AtlantECO climatology, the annual mean FFGM biomass fields and the statistics (Table 4) are calculated from these "model-sampled" biomasses to avoid bias due to sampling.
For other variables, model outputs ($NO_3^-$, $PO_4^{3-}$, Chl, Mesozooplankton, GM+FFGM) were regridded horizontally and vertically on the same grid as the corresponding observations (see previous section). The macrozooplankton and mesozooplankton fields were integrated vertically on the appropriate vertical range. When compared to observations, model outputs are sampled at exactly the same location and month as the observations. Annually averaged fields as well as statistics (Table 4) are computed from these sampled fields to avoid bias due to sampling.





**3 Results**

**3.1 Macrozooplankton biomass**

**3.1.1 Simulated biomass**

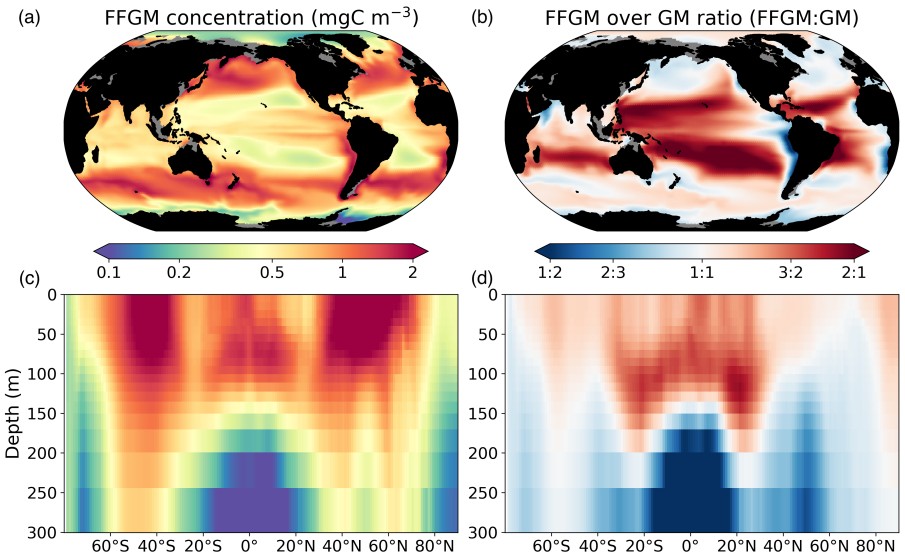

**Figure 3. FFGM and FFGM:GM ratio.** Annual mean of FFGM carbon concentrations (mg C m$^{-3}$, log-scale), averaged over the top 300 meters (a), and zonally averaged (c). Annual of mean FFGM:GM ratio, averaged over the top 300 meters (b), and zonally averaged (d). Red tones indicate FFGM dominance, blue tones indicate GM dominance.

We first analyse the simulated living compartments of PISCES-FFGM. The total integrated biomass of all living compartments simulated by PISCES-FFGM is 1.4 Pg C for the upper 300 meters of the global ocean. Primary producers account for
51% of this biomass. Total macrozooplankton accounts for 12% of the total biomass. Our model predicts that FFGM and GM contribute roughly equally to macrozooplankton biomass, each having a biomass of about 0.08 Pg C. Figure 3 displays the annual mean FFGM carbon concentration and FFGM to GM ratio averaged over the top 300 m of the ocean. It also shows the zonally averaged distribution of this concentration and of this ratio. Simulated FFGM concentration is high (>1 mg C m$^{-3}$) in the subpolar regions and close to the equator and low (<1 mg C m$^{-3}$) in the oligotrophic gyres and at extreme latitudes.
The most striking feature is the reverse distribution of the ratio as compared to the simulated absolute biomass of both GM and FFGM. The ratio exceeds 2 in oligotrophic subtropical gyres while it is minimal in the most productive regions. In eastern boundary upwelling systems, FFGM biomass can be more than two times lower than GM biomass. Vertically, the ratio is on average larger than 1 in the euphotic zone. Below the euphotic zone, it sharply decreases as GM become dominant. In the mesopelagic domain, flux-feeding has been shown to be a very efficient mode of predation (Jackson, 1993 ; Stukel, Ohman, et





 al., 2019). Since FFGM are not able to practice this feeding mode, they are outcompeted by GM. FFGM:GM ratio is maximum in the lower part of the euphotic zone in the subtropical domain where deep chlorophyll maxima are located.

### 3.1.2 Comparison to observations

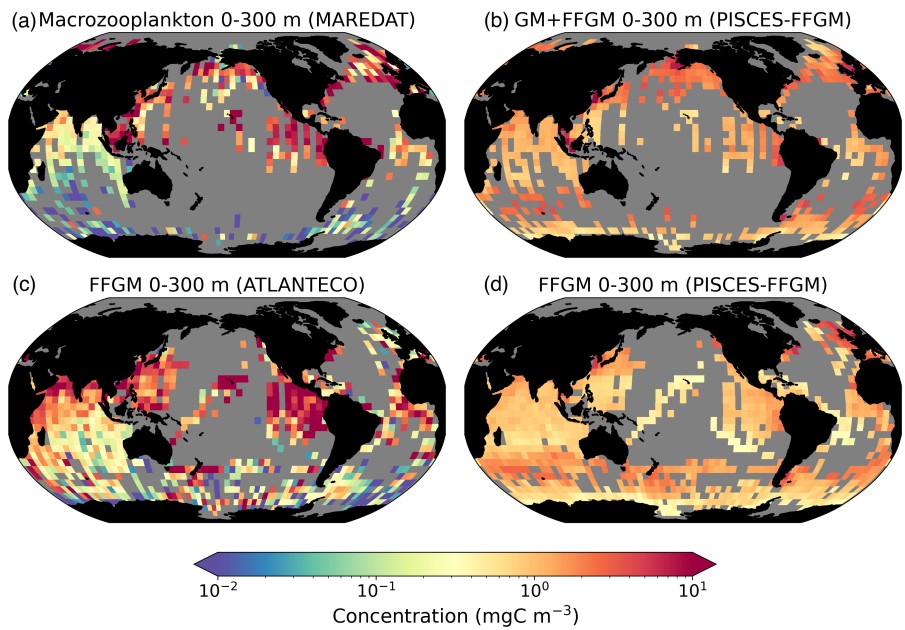

**Figure 4. Comparison between observed and modeled macrozooplankton biomasses.** Annual means of carbon concentrations (mg C m$^{-3}$, log-scale), averaged over the top 300 meters on a 5°resolution grid. (a) macrozooplankton from MAREDAT (b) "model-sampled" total macrozooplankton (GM+FFGM) (c) FFGM from AtlantECO climatology (d) "model-sampled" FFGM. As described in section 2.4.3, modeled biomasses were sampled where observations were available.

Then, we focus on the evaluation of the new components added in this version of PISCES, i.e. GM and FFGM. In the appendices, we present an evaluation of nitrate, chlorophyll and mesozooplankton (See Text A2 and Fig. A2). For these tracers, note that the performance of PISCES-FFGM is similar to that of PISCES-v2 (Aumont et al., 2015).

The annual mean distributions of total macrozooplankton (FFGM and GM) and FFGM only, averaged over the top 300 m of the ocean, are compared to available observations (Figure 4). A quantitative statistical evaluation of the model performance for these two fields is presented in Table 4. The Spearman correlation coefficient between observed and modeled total macrozooplankton biomasses is 0.26 (p-value < 0.001). Regions of high macrozooplankton biomass are correctly simulated in the northern hemisphere by our model: 94% of the area in which observed concentrations are greater than 0.5 mg C m$^{-3}$ correspond to areas in which the simulated concentration is greater than 0.5 mg C m$^{-3}$. On the other hand, observations suggest moderate biomass in the Indian Ocean (between 0.05 and 0.5 mg C m$^{-3}$) and low biomass in the Southern Ocean (lower than 0.05 mg C m$^{-3}$). These low and moderate biomasses are not captured by our model, which simulates values greater than 0.5




|  |  | Total Macrozooplankton | FFGM | FFGM | FFGM | FFGM |
| --- | --- | --- | --- | --- | --- | --- |
|  | Experiment | PISCES-FFGM | PISCES-FFGM | PISCES-CLG | PISCES-HGR | PISCES-HGM |
| Model | Mean (mg C m$^{-3}$) | 1.65 | 1.18 | 0.69 | 5.02 | 1.24 |
|  | Median (mg C m$^{-3}$) | 1.56 | 0.80 | 0.30 | 4.59 | 0.98 |
|  | Std (mg C m$^{-3}$) | 1.29 | 0.96 | 0.69 | 3.00 | 0.86 |
| Observation | Mean (mg C m$^{-3}$) | 11.01 | 8.22 | 8.22 | 8.22 | 8.22 |
|  | Median (mg C m$^{-3}$) | 0.52 | 1.11 | 1.11 | 1.11 | 1.11 |
|  | Std (mg C m$^{-3}$) | 128 | 26.9 | 26.9 | 26.9 | 26.9 |
| comparison | Bias (mg C m$^{-3}$) | -9.36 | -7.04 | -7.53 | -3.20 | 6.98 |
|  | Bias (log10) | 0.57 | 0.04 | -0.18 | 0.60 | 0.02 |
|  | R Spearman | 0.26 ($p < 10^{-5}$) | 0.17 ($p < 10^{-5}$) | 0.34 ($p < 10^{-5}$) | -0.28 ($p < 10^{-5}$) | -0.22 ($p < 10^{-5}$) |
|  | High biomasses match | 94 % | 91 % | 84 % | 100% | 85% |
|  | Low biomasses match | 2 % | 14 % | 41 % | 0% | 18% |

**Table 4. Macrozooplankton model vs. observation statistics.** "Mean", "median" and "standard" deviation are computed on all the non-zero biomass values of the annual climatologies (as defined in section 2.4.3 of the methods) weighted by their respective cell areas. "Bias" is computed as the difference between modeled and observed means. "Bias (log10)" is computed on log10 converted observed and modeled climatologies. "R Spearman" is the Spearman correlation coefficient computed on non zero values of the climatologies. "High biomasses match" is the percentage of observed area where biomasses are greater than 0.5 mg C m$^{-3}$ that correspond to area where model biomasses are greater than 0.5 mg C m$^{-3}$. "Low biomasses match" is the percentage of observed area where biomasses are lower than 0.5 mg C m$^{-3}$ that correspond to area where model biomasses are lower than 0.5 mg C m$^{-3}$.

mg C m$^{-3}$ in both areas: 98% of the area in which observed concentrations are lower than 0.5 mg C m$^{-3}$ correspond to areas
in which modeled concentrations are greater than 0.5 mg C m$^{-3}$. Overall, the simulated distribution of macrozooplankton is too homogeneous with respect to what the observations suggest. This is confirmed by the much smaller standard deviation in our model simulation than in the observations, 1.3 and 128 mg C m$^{-3}$ respectively.

Our model simulates a distribution of FFGM in the upper ocean that correlates with observation with a Spearman correlation coefficient of 0.17 (p-value < 0.001). The simulated FFGM biomass is high (>0.5 mg C m$^{-3}$) in the equatorial domain of the
Pacific and Atlantic oceans and in the mid latitudes of both hemispheres. Conversely, FFGM biomass is moderate (between 0.05 and 0.5 mg C m$^{-3}$) in the oligotrophic subtropical gyres and in the high latitudes (>60°). Compared to observations, the spatial patterns of high biomasses are better reproduced than for total macrozooplankton: 91% of the area in which observed concentrations are greater than 0.5 mg C m$^{-3}$ correspond to areas in which modeled concentrations are greater than 0.5 mg C m$^{-3}$. However, the maximum observed values are strongly underestimated: the 95th percentile of the modeled values is 2.6 mg
C m$^{-3}$ while it is 32 mg C m$^{-3}$ in the observations. In the Southern Ocean, the simulated distribution is much more zonally homogeneous than suggested by observations (Fig. 4). Overall, the predicted median biomass of FFGM is similar to that of observations, 0.80 vs. 1.11 mg C m$^{-3}$. As with macrozooplankton, but to a lesser extent, the simulated standard deviation is





significantly lower than in the observations, 0.96 and 26.9 mg C m$^{-3}$ respectively. The standard and log10 biases are closer to 0 than those calculated for macrozooplankton (Table 4).

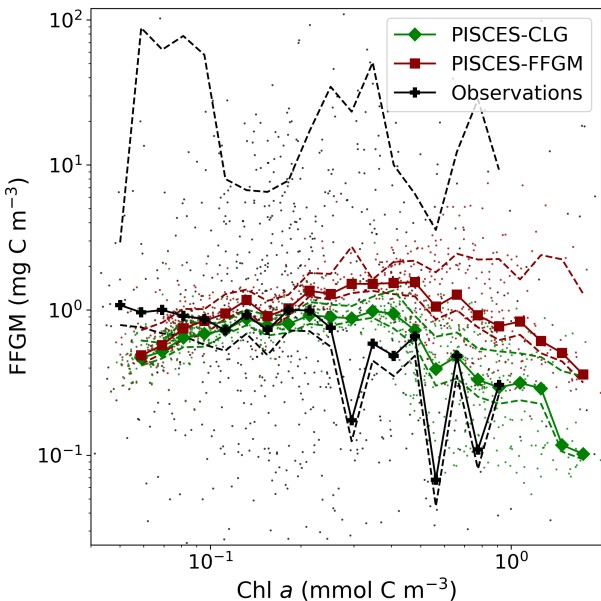

**Figure 5. Chlorophyll-FFGM relationship.** Log-log scatter plot showing FFGM concentration versus total chlorophyll concentration for PISCES-FFGM, PISCES-CLG clogging run, and for the AtlantECO vs OC-CCI chlorophyll datasets. The datasets were gridded into an annual climatology with a spatial resolution of 5°. Each small dot corresponds to one grid cell of these climatologies. Large dots connected by a line represent the median per 0.07-wide log-bins of chlorophyll, dashed lines represent standard deviations below and above the median for each bin.

We also compared the observed and modeled relationships between FFGM biomass distributions and chlorophyll levels. Black dotted line and points on figure 5 show the FFGM biomass from the AtlantECO database plotted against the corresponding chlorophyll concentrations from OC-CCI (see section 2.4.2). Despite considerable scatter, this data-based analysis suggests a modest decrease of FFGM biomass for chlorophyll concentrations above about 0.3 mg Chl m$^{-3}$. Yet, this decrease is far from systematic, since even at high chlorophyll concentrations, FFGM biomass can be very high (>10 mg Chl m$^{-3}$). In

our reference PISCES-FFGM simulation (red dotted-line and points on figure 5), the median values of FFGM biomass appear to be consistent with observations at intermediate chlorophyll concentrations between 0.08 and 0.3 mg Chl m$^{-3}$. However, as already mentioned in the previous section, our model predicts a much weaker variability of FFGM biomass. For higher chlorophyll concentrations, median FFGM levels become significantly larger than in the observations (up to one order of magnitude larger, see fig. 5).





### 3.1.3 Sensitivity experiments

Here, we present the PISCES-HGR, PISCES-HGM and PISCES-CLG sensitivity experiments and their influence on the FFGM modeled distributions.

A 5-fold increase in the maximum growth rate in the PISCES-HGR experiment leads to a 4-fold and 5-fold increase in the mean and median FFGM concentrations, respectively (see Table 4). While the mean is closer to the observed mean than in the standard experiment, the negative Spearman coefficient shows the unrealistic nature of this simulation and the need to correct mortality accordingly (see Table 4 and Fig. A3). The increase in mortality rate in the PISCES-HGM experiment results in similar mean and median FFGM biomass to the standard PISCES-FFGM experiment (see Table 4) and Fig. A4) but a worse data-model fit (see Table 4 and Fig. A6). Given the large range suggested for the growth rate of FFGM (0.105-1.85 $d^{-1}$ according to Luo et al. (2022)), these results supports the choice of a conservative approach in our reference experiment (PISCES-FFGM) where the FFGM maximal growth rate is identical to that of GM (i.e. 0.28 $d^{-1}$).

The addition of clogging in PISCES-CLG increases the model-data spatial correlation (Spearman's correlation coefficient is 0.34 compared to 0.17 previously, see Table 4 and Fig. A6). This improvement is explained by a better representation of areas with moderate and low biomass in PISCES-CLG (concentrations <0.5 mg $m^{-3}$), especially in the southern part of the Southern Ocean (see Fig. A5). Indeed, 41% of the areas where observations give values below 0.5 mg C $m^{-3}$ correspond to areas where the model predicts values below 0.5 mg C $m^{-3}$ (vs. only 14% in PISCES-FFGM). Also, as shown in Fig. 5, the addition of clogging (green dotted line and points) reduces the bias and thus reproduces the observed relationship between FFGM biomass and chlorophyll *a* concentration better than the standard experiment. However, the simulated spatial variability remains strongly underestimated (std = 0.69 mg C $m^{-3}$ in PISCES-CLG and 26.9 mg C $m^{-3}$ in the AtlantECO climatology) and biases are increased when clogging is added (see Table 4).

None of the sensitivity experiment reproduce the observed spatial variability, which remains much higher than the modeled spatial variability similarly to the standard experiment, and the distribution of observed biomasses is consequently much more spread out than the model (see Fig. A6).

## 3.2 Carbon cycle

### 3.2.1 Carbon export from the surface ocean

We first discuss the role of macrozooplankton in shaping the carbon cycle in the upper ocean, focusing on differences between GM and FFGM-related surface processes. Table 5 shows the globally integrated sinking flux of organic carbon particles at 100 m and 1000 m, while Figure 6 focuses on the FFGM-driven carbon fluxes. The total export flux from the upper ocean (at 100 m) is 7.55 Pg C $yr^{-1}$ (Table 5). This value is relatively similar to previous estimates using different versions of PISCES (Aumont et al., 2015, 2017, 2018). It is also within the range of published estimates, *i.e.* 4-12 Pg C $yr^{-1}$ (e.g., Laws et al., 2000; Dunne et al., 2007; Henson et al., 2011; DeVries and Weber, 2017). Small and large particles produced by phytoplankton, microzooplankton and mesozooplankton account for 91% of this carbon flux. The remaining 9% (0.69 Pg C $yr^{-1}$, Table 5) is due to macrozooplankton (FFGM+GM), with one third of this amount coming from carcasses and the remaining from fecal



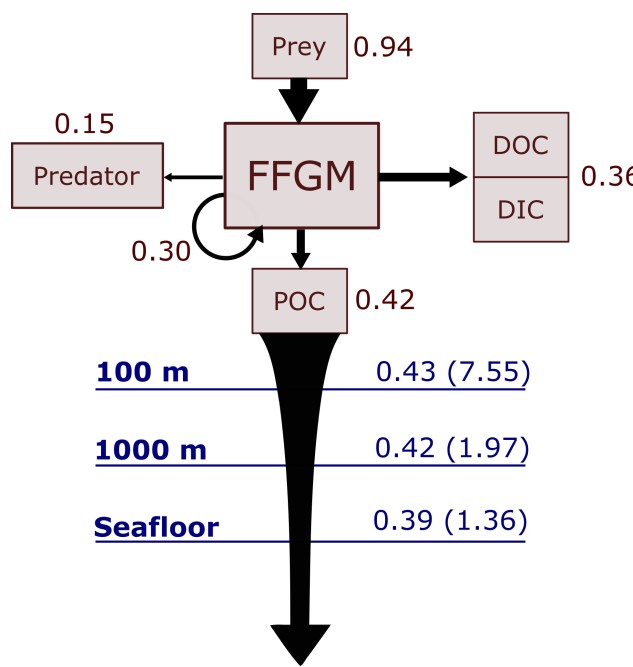

**Figure 6. Schematic representation of carbon fluxes induced by processes related to FFGM.** Values are in Pg C yr$^{-1}$. The upper part of the diagram represents the sources and sinks of FFGM integrated globally over the first 100 meters. The source is the grazing on the different prey. The arrow going from FFGM to FFGM corresponds to the flux related to growth due to assimilated food. The sinks are : i) the remineralization, non-assimilation and linear mortality that go into the dissolved organic carbon (DOC) and dissolved inorganic carbon (DIC) ii) the quadratic predatory mortality term (directly remineralized in PISCES-FFGM because of the lack of explicit representation of upper level predators) and iii) the production of particular organic carbon (POC) via carcasses and fecal pellets. The lower part of the diagram corresponds to the export of POC linked to the fall of carcasses and fecal pellets of FFGM. The values in blue correspond to the global annual FFGM-driven POC flux through the corresponding depth, the values in parenthesis representing the total POC flux (i.e. related to FFGM, GM, bPOC and sPOC).

pellets. FFGM are responsible for an export of 0.43 Pg C yr$^{-1}$ (Table 5), which represents 62% of the total macrozooplankton contribution.

The particularly large contribution from FFGM compared to GM comes from higher production (grazing of 0.94 Pg C yr$^{-1}$ compared to 0.63 Pg C yr$^{-1}$ for GM, Fig. 6 and S7) while both groups shows similar export efficiency. 45% of the grazed matter is exported at 100 m, the remaining 55% is split between implicit predation by upper trophic levels and loss to dissolved inorganic and organic carbon.

### 3.2.2    Carbon transfer efficiency to the deep ocean

We then analyze how the representation of the two new macrozooplankton groups influences the fate of particulate organic carbon in the deep ocean. At 1000 m, the total simulated POC flux is 1.97 Pg C yr$^{-1}$ (Table 5). Thus, the flux transfer efficiency





| Experiment | Depth (m) | bPOC (Pg C yr$^{-1}$) | sPOC (Pg C yr$^{-1}$) | $Fp_{GM}$ (Pg C yr$^{-1}$) | $Ca_{GM}$ (Pg C yr$^{-1}$) | $Fp_{FFGM}$ (Pg C yr$^{-1}$) | $Ca_{FFGM}$ (Pg C yr$^{-1}$) | Total (Pg C yr$^{-1}$) | GM+FFGM contribution | FFGM contribution |
|---|---|---|---|---|---|---|---|---|---|---|
| PISCES-FFGM | 100 | 4.49 | 2.37 | 0.09 | 0.17 | 0.29 | 0.14 | **7.55** | 9% | 6% |
| PISCES-GM | 100 | 4.92 | 2.49 | 0.11 | 0.20 | 0.00 | 0.00 | **7.73** | 4% | 0% |
| PISCES-LOWV | 100 | 4.72 | 2.41 | 0.08 | 0.15 | 0.24 | 0.12 | **7.71** | 8% | 5% |
| PISCES-FFGM | 1000 | 1.18 | 0.12 | 0.11 | 0.14 | 0.27 | 0.15 | **1.97** | 34% | 21% |
| PISCES-GM | 1000 | 1.27 | 0.13 | 0.12 | 0.16 | 0.00 | 0.00 | **1.68** | 17% | 0% |
| PISCES-LOWV | 1000 | 1.23 | 0.13 | 0.04 | 0.06 | 0.07 | 0.04 | **1.56** | 13% | 7% |

**Table 5. Particulate carbon flux composition at 100 and 1000 m.** Units are in Pg C yr$^{-1}$. sPOC (resp. bPOC) is for small (resp. large) particulate organic carbon. $Ca_{GM}$ (resp. $Ca_{FFGM}$) is for GM (resp. FFGM) carcasses. $Fp_{GM}$ (resp. $Fp_{FFGM}$) is for GM (resp. FFGM) fecal pellets.

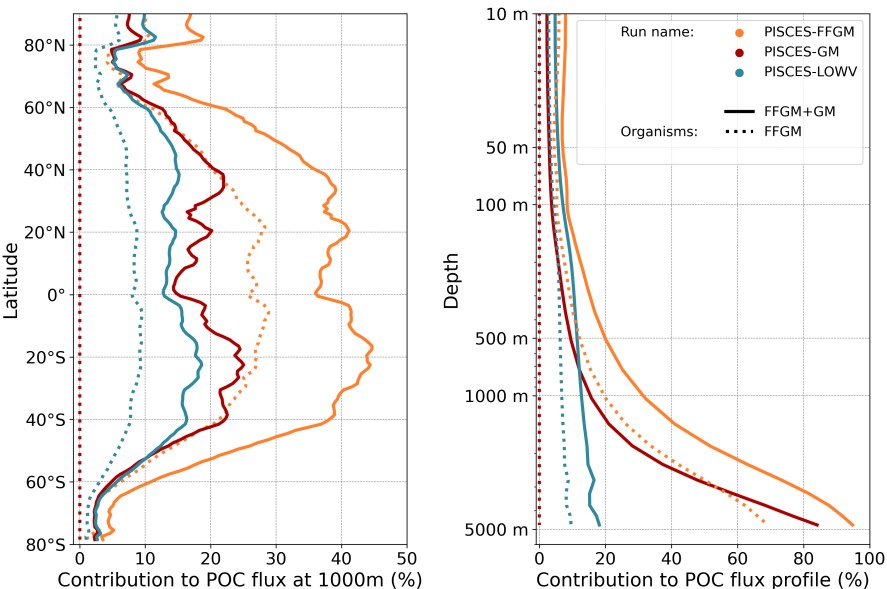

**Figure 7. Macrozooplankton relative contribution to particulate organic carbon fluxes.** The color indicates the PISCES configuration considered (see sensitivity section). The figure on the left shows the relative contribution of FFGM (dash) and macrozooplankton (FFGM+GM, solid) to the POC export at 1000 m averaged zonally. The figure on the right shows the globally averaged vertical profile of these relative contributions.

from 100 m to 1000 m is 26%. Most of this strong flux reduction is due to the loss of small and large organic particles. Macrozooplankton-driven export is very effective because it remains almost unchanged from 100 m to 1000 m, 0.69 and 0.67 Pg C yr$^{-1}$, respectively (Table 5). Therefore, the contribution of macrozooplankton increases strongly with depth to 34% of

the total carbon export at 1000 m (Fig. 7). The respective contribution of particles produced by GM and FFGM (carcasses and



fecal pellets) to this flux is almost identical at both depth horizons. At 5000 m, more than 90% of the carbon flux is due to macrozooplankton (Fig. 7).

The PISCES-LOWV sensitivity experiment, in which carcasses and fecal pellets sinking speeds of both macrozooplankton groups are reduced to 30 m d$^{-1}$, shows a much greater attenuation of POC fluxes with depth: while the total export of organic
carbon at 100 m increases slightly to 7.71 Pg C yr$^{-1}$, it is reduced by 20% at 1000 m compared to the standard PISCES-FFGM run (1.56 Pg C yr$^{-1}$, see table 5). The macrozooplankton contribution is similar to that found in the standard model at 100 m (8%) but the contribution is reduced to 13% at 1000 m and to 20% at 5000 m (Fig. 7). This confirms that the strong contribution of macrozooplankton to POC fluxes at depth in the standard run is explained by the very high sinking speeds of carcasses and fecal pellets. These high sinking speeds prevent any significant remineralization of these particles as they sink to the seafloor.

The PISCES-GM sensitivity experiment, in which FFGM are not allowed to grow, shows a similar depth gradient of the macrozooplankton contribution (Fig. 7, red curve) compared to the standard run, but a lower contribution at each depth (by 10%). Indeed, the transfer efficiency from 100 to 1000 m differs by only 2% between the two groups in the standard model (97% for FFGM, 95% for GM) so that particles produced at the surface by both groups have a similar fate towards the deep ocean. However, the estimated transfer efficiency is biased as both groups of organisms produce particles below 100 m. Because
they can adopt a flux feeding strategy of predation, GM occupy the whole water column whereas FFGM remain confined to the upper ocean (see section 3.1 and Figure 3). As a result, GM also produce particles below 100 m which contribute to the flux at 1000 m and explains the computed higher transfer efficiency. This is confirmed by the PISCES-LOWV experiment: the efficiency of FFGM is reduced to 30% in this simulation while that of GM is only reduced to 40%, even though the carcasses and fecal pellets sinking velocities of both groups are identical. As the remineralization processes are identical in the two runs,
we can reasonably assume that the difference comes from the relatively higher productivity below 100 m of GM compared to FFGM.

### 3.2.3 POC flux spatial patterns

Although the processes underlying the efficient sequestration of the particulate carbon issued from the two groups of macrozooplankton are similar, we investigate how the spatial and temporal patterns of the induced deep POC export differ between
GM and FFGM.

The relative contribution of FFGM and GM to the POC flux at 1000 m presented in Figure 8 is very contrasted between the two macrozooplankton groups. The POC flux due to FFGM is maximal at about 40% of the total flux in the oligotrophic subtropical gyres. In the productive areas of the low and mid-latitudes, it has intermediate values close to 25%. It is minimal (<15%) at high latitudes, especially along the Antarctic. In contrast, POC fluxes due to GM are maximal in the productive
regions of the low and mid-latitudes, especially in boundary upwelling systems where they can exceed 35% of the total flux. These patterns are consistent with the respective spatial distribution of FFGM and GM (ratio shown in figure 3).

We further investigate the importance of GM and FFGM for the spatial patterns of the export of carbon to the deep ocean by contrasting PISCES-FFGM and PISCES-GM experiments (see Section 2.3). Figure 7 shows the relative contribution of macrozooplankton to POC flux as a function of latitude. By comparing the standard model (orange curve) with the experiment





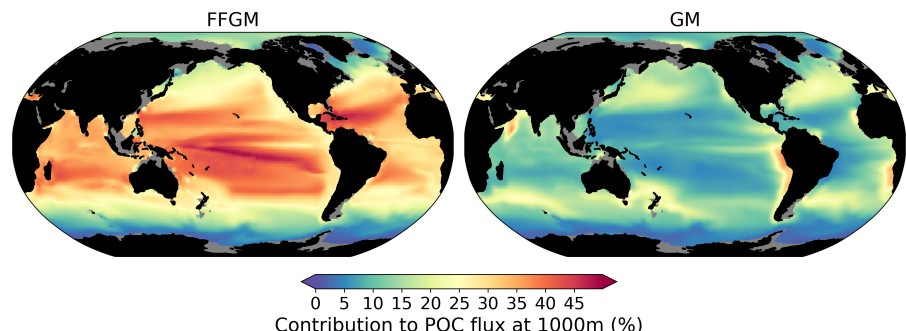

**Figure 8. Relative contribution of macrozooplankton to particulate organic carbon flux at 1000 m.** On the left (resp. right): relative importance at 1000 m of FFGM (resp. GM) carcasses and fecal pellets driven POC flux to total POC flux (incl. GM and FFGM carcasses and fecal pellets as well as small and large particles).

without FFGM (PISCES-GM, red curve), we deduce that the explicit representation of FFGM alters strongly the latitudinal distribution of this relative contribution. It is significantly increased at all latitudes. This increase is particularly important in the low latitudes where the contribution goes from less than 20% when FFGM are not allowed to grow (PISCES-GM) to more than 45% in the reference simulation PISCES-FFGM. Furthermore, export due to GM is maximal at about 20°N and S. Compared to GM, the FFGM contribution is relatively constant between these latitudes. This result highlights the strong efficiency of

FFGM at exporting organic matter to the deep ocean, in particular in oligotrophic regions with low productivity. The addition of FFGM reduces the contribution of GM at all latitudes, especially at mid and low latitudes in which the contribution losses 15 to 20% (Fig. 7). This reduction results from the competition between FFGM and GM.

## 4  Discussion

We added an explicit representation of two macrozooplankton groups in PISCES-FFGM: a generic macrozooplankton group,

for which the parameterization is based on an allometric scaling of the mesozooplankton group already existing in PISCES-v2 (Aumont et al. (2015), see section 2.2) and which feed mainly on the latter, and an FFGM group that can feed on phytoplankton and microzooplankton. The introduction of FFGM into PISCES, based solely on the representation of their specific diet due to the filter-feeding mode, provided some insights into the potential impacts of FFGM on planktonic communities and carbon cycling at the global scale through trophic effects (e.g. competition with generic macrozooplankton) and efficient carbon export.

### 4.1  Comparison to previous modeling studies

#### 4.1.1  Macrozooplankton biomass

After the addition of FFGM in PISCES, our simulation results consistently show that FFGM dominate macrozooplankton in low-productivity regions, but that absolute abundances of FFGM are nonetheless higher in productive areas of the world ocean



(Fig. 3). In a recent study using the COBALTv2 biogeochemical model, Luo et al. (2022) explored the role of pelagic tunicates

in the marine ecosystem, with the addition of two new plankton functional groups, i.e. a large salp/doliolid group similar to our FFGM, and a small appendicularian group (Luo et al., 2022). They showed that the FFGM:GM ratio in their model follows a decreasing relationship with chlorophyll, consistently with our modeled FFGM:GM ratio patterns. To better reproduce the relationship between AtlantECO FFGM biomass and chlorophyll from the OC-CCI product, the addition of clogging was needed in our model (Fig. 5 and section 3.1). Given the paucity of data, it is currently difficult to evaluate these model insights

from macrozooplankton databases alone. Heneghan et al. (2020) showed that salps dominate other macrozooplankton groups in low-productivity regions, but, contrary to our model results, these authors also showed that these organisms are more abundant in absolute values in these low-productivity regions than elsewhere in the ocean. Yet, they did not explore the processes that could drive this distribution. As evidenced by our PISCES-CLG experiment, clogging may be a potential explanatory mechanism but the evidence for this process is weak. Future studies are needed to determine the processes involved in limiting

FFGM biomass at high chlorophyll concentrations.

### 4.1.2   Export of organic carbon

Our modeled FFGM have a weak impact on phytoplankton and microzooplankton biomasses, due to the low predation pressure they exert on these low-trophic levels (grazing flux of 1 Pg C yr$^{-1}$, which represents less than 3% of primary productivity). Nevertheless, due to the high sinking speed of FFGM-derived fecal pellets and carcasses, FFGM substantially increase the

carbon export ratio and transfer efficiency. We compiled results from distinct studies on global biogeochemical impacts of FFGM in table 6 to support our results.

The overall PISCES-FFGM modeled production of POC by FFGM in the upper 100 m is 0.42 Pg C yr$^{-1}$ (Table 6). This value falls within the range of data-driven estimates (Table 6). It is an order of magnitude above the value of 0.03 Pg C yr$^{-1}$ from Lebrato et al. (2019), presented as a lower bound estimate due to their conservative assumption of equivalence between

GZ annual production and total GZ biomass. On the other hand, our simulated FFGM POC production within the top 100 m is 10 times lower than the estimate of 3.9 Pg C by Luo et al. (2020). In this study, FFGM production was forced offline by modeled phytoplankton and zooplankton climatologies, so that FFGM predation had no feedback on their prey biomass. Luo et al. (2020)'s production estimate can be seen as an upper estimate as GZ-induced predation pressure would affect the biomass of other trophic levels in a fully-coupled model, thus affecting the gelatinous biomass itself and the induced carbon

fluxes. Indeed the higher FFGM POC production is mostly due to a higher FFGM grazing in their study (6.6 Pg C yr$^{-1}$ compared to our modeled value of 1 Pg C yr$^{-1}$, Table 6). Finally, our modeled FFGM impacts on upper ocean POC are similar to those by Luo et al. (2022) based on COBALT-GZ: the simulated production of detritus by FFGM in the first 100 m in our model is twice lower than in Luo et al. (2022) and the effective export of these detritus at 100 m is 30% lower (Table 6). The smaller difference in export than in production lies in the use of a 10-times lower particle sinking speed and a 20-times higher

remineralization rate in COBALT-GZ (Stock et al., 2014) compared to PISCES-FFGM, resulting in a lower production export efficiency in COBALT-GZ than in PISCES-FFGM (Table 6). Note that appendicularians in GZ-COBALT produced 4 times less





| | Source | | PISCES-FFGM | (Luo et al., 2022) | (Luo et al., 2020) | (Luo et al., 2020) | PISCES-LOV | (Lebrato et al., 2019) | (Henschke et al., 2016) |
|---|---|---|---|---|---|---|---|---|---|
| | Type of study | | model | model | data-driven | data-driven | model | data-driven | data-driven |
| | Ca_FFGM sinking speed | m d$^{-1}$ | 800 | 100 | 1000 | 800 | 30 | 800-1200 | 0–1700 |
| | Fp_FFGM sinking speed | m d$^{-1}$ | 1000 | 100 | 650 | 100 | 30 | 800-1200 | 490–4000 |
| Biomasses | Vertically integrated biomass | TgC | 133 | 102 ⋆ | - | - | - | - | - |
| | Upper 100 m biomass | TgC | 48.5 | 81.5 | - | - | - | - | - |
| Surface Ocean POC export | Total grazing by FFGM | Pg C yr$^{-1}$ | 0.94 | - | 6.6 | - | - | - | - |
| | Predation on FFGM by UTL | Pg C yr$^{-1}$ | 0.15 | 0.1 | 0.94 | - | - | - | - |
| | FFGM POC Prod. top 100 m | Pg C yr$^{-1}$ | 0.42 | 0.79 | 3.91 | 3.91 | 0.44 | <0.04* | - |
| | Ca_FFGM contrib. to POC | % | 35% | 20% | 20% | - | - | - | - |
| | Fp_FFGM contrib. to POC | % | 65% | 80% | 80% | - | - | - | - |
| | FFGM driven POC exp. 100 m | Pg C yr$^{-1}$ | 0.43 | 0.57 | 2.7 | 1.3 | 0.36 | - | - |
| | FFGM export efficiency | % | 100% | 72% | 69% | 33% | 82% | - | - |
| | FFGM contrib. to POC100 | % | 6% | 9% | 20% | 10% | 5% | - | - |
| | Dif. in POC100 (with vs without FFGM †) | % | -2% | +2% | - | - | - | - | - |
| | Dif. in tot MAC contrib. to POC100 (with vs without FFGM †) | % | +55% | +41% | - | - | - | - | - |
| | Dif. in GM contrib. to POC100 (with vs without FFGM †) | % | -19% | -11% | - | - | - | - | - |
| Deep Ocean POC export | FFGM driven POC exp. 1000 m | Pg C yr$^{-1}$ | 0.42 | - | 1.4 | 0.33 | 0.11 | <0.02-0.03* | - |
| | FFGM driven POC exp. Seafloor | Pg C yr$^{-1}$ | 0.39 | - | 0.86 | 0.17 | 0.002 | <0.01* | - |
| | FFGM POC Teff 100 m to 1000 m | % | 97% | - | 52% | 25% | 30% | 46-54% | - |
| | Yearly max. FFGM POC exp. ‡ | mg C m-2 | 141 (min : 0.34 , max : 1580) | | - | - | 38 (min : 0.30 , max : 323) | - | 128 - 6725 (min : 0.6 - 1171 , max : 656 - 77 143) |

**Table 6. Comparison of parameters related to the impact of FFGM on the carbon cycle between different global scale studies based on data and/or models.** Ca_FFGM is for FFGM carcasses. Fp_FFGM is for FFGM fecal pellets. UTL is for Upper Trophic Levels. POC is for Particulate Organic carbon. Prod. is for Production. Contrib. is for contribution. Dif. is for Difference. Export efficiency is the ratio between the POC export below 100 m and the POC production in the upper 100 m. POC100 is for total POC export below 100 m. exp. is for export to. Teff is for transfer efficiency. Tot MAC is for total macrozooplankton (GM + FFGM). * Lebrato et al. (2019) consider also cnidarians and ctenophores. ⋆ Luo et al. (2022) integrate FFGM biomass includes appendicularians. † We assume that our comparison between PISCES-FFGM and PISCES-GM is consistent with Luo et al. (2022)'s comparison between GZ-COBALT and COBALTv2. ‡ (Henschke et al., 2016) provides an estimate of POC export at 1000 m during a localized 1-month duration swarm event, the range is based on the spread of the results considering different species. We compare those values to the yearly maximum FFGM-driven POC export at 1000 m in our model, the range is based on the spread of the results considering all different grid cells.





detritus in the upper 100 m than large tunicates, which supports our choice to represent only FFGM (i.e. macrozooplankton) and not filter-feeding mesozooplankton in our biogeochemical model.

The impact of an explicit representation of FFGM on POC export is negligible in both models when compared to a version
without FFGM (+/- 2%, Table 6). But the contribution of total macrozooplankton to POC fluxes increases significantly with FFGM in both models (GZ-COBALT: +41%, PISCES-FFGM: +55%, Table 6) and this despite the simulated decrease in export by GM (-11% in GZ-COBALT, -19% in PISCES-FFGM, Table 6), so that the contribution of FFGM only to POC export at 100 m in both models is more than 5% (Table 6). Thus, we can reasonably state that the representation of FFGM in a biogeochemical model redistributes the carbon particles between the different compartments over the top 100 m (more of
very large particles from macrozooplankton, less of small particles from smaller organisms) without significantly altering the total amount. This change in particles composition is key to the major role that FFGM play in the export of carbon to the deep ocean.

### 4.1.3 Deep carbon fluxes

FFGM have a modest impact on subsurface export (less than 10 % of the global POC export at 100 m depth), but this impact is
highly increasing with depth, reaching much higher values at the seafloor (>40%) and suggesting that FFGM play a key role in carbon sequestration in the deep ocean. We also demonstrated that surface FFGM productivity and the transfer efficiency of FFGM-driven POC are key processes that strongly affect the magnitude and distribution of deep POC export.

The FFGM-driven export of POC at 1000 m (resp. seafloor) of 0.42 (resp. 0.39) Pg C yr$^{-1}$ falls between the low value of 0.02 (resp. 0.01) Pg C yr$^{-1}$) proposed by Lebrato et al. (2019) and the much larger estimate of 1.4 (resp. 0.86) Pg C yr$^{-1}$ given
by Luo et al. (2020) (Table 6). The quite large differences between these estimates are mainly explained by the evaluation of surface FFGM productivity: FFGM productivity is 10 times higher in Luo et al. (2020) than in our study. In contrast, Lebrato et al. (2019) used for gelatinous zooplankton a biomass estimate of 38 TgC provided by Lucas et al. (2014), which resulted in low export values (<0.04 Pg C yr$^{-1}$) at all levels of the water column.

In addition to surface productivity, the efficiency of POC transfer is critical to the absolute value of POC export at depth.
The sinking velocity of particles is a key factor that strongly controls this efficiency. In the studies of Lebrato et al. (2019) and Luo et al. (2020), in which the sinking velocities are greater than 650 m d$^{-1}$, the transfer efficiency is about 50% (Table 6). It is reduced to 25% when the FFGM fecal pellets (which account for 80% of FFGM detritus in their study) velocity is reduced to 100 m d$^{-1}$ in Luo et al. (2020). The same finding was obtained when reducing the velocity from 800-1000 m d$^{-1}$ to 30 m d$^{-1}$ in our experiment PISCES-LOWV, where the transfer efficiency from 100 to 1000 m decreases from 97% to 30%. However,
due to the use of a low remineralization rate, our simulated transfer efficiency from 100 to 1000 m is very high compared to Luo et al. (2020) for similar carcasses and fecal pellets sinking speeds (Table 6). Still, our transfer efficiency in PISCES-FFGM fits the vertical profiles of depth attenuation of jelly-driven organic matter export proposed by Lebrato et al. (2011) for high sinking velocities and low remineralization rates.

Last but not least, PISCES-FFGM seems to capture the intensity and part of the variability of the intense carbon export events
described by Henschke et al. (2016) linked to short time proliferation events of FFGM: they estimated the export potential at





1000 m of different salps species during a 1 month swarm. Mean values ranged from 128 to 6725 mg C m$^{-2}$ depending on the species, the minimum from 0.6 to 1171 mg C m$^{-2}$ and the maximum from 656 to 77 143 mg C m$^{-2}$. We compare these results to the annual maxima of the FFGM carbon export simulated at each grid point by our model (Table 6). The values obtained range from 0.34 to 1580 mg C m$^{-2}$ with a spatial mean of 141 mg C m$^{-2}$, which is consistent with the species-range of mean,
min and max in their study (Table 6). This also supports our choice of a very low remineralization rate and high sinking rates. The latter is confirmed with the PISCES-LOWV experiments in which modeled export maxima fall below the min, mean and max ranges of Henschke et al. (2016).

## 4.2 Data-based climatology

To evaluate the modeled FFGM biomasses, we compiled data from different sources (section 2.4) to produce a gridded clima-
tology of large pelagic tunicates. Our AtlantECO dataset is based on similar observations as the previously compiled dataset (Luo et al., 2020, 2022), but we used a different approach to convert abundances to biomasses by taking into account the taxonomic information available on the samples, even when the species is not indicated.

Our model predicts a median biomass of FFGM similar to our dataset (0.80 vs. 1.11 mg C m$^{-3}$), and reproduces 91% of the areas where biomass is high (>0.5) (Table 4). The introduction of a clogging mechanism, which would represent a saturation
of the salp filtering apparatus for high prey concentrations, improves the representation of low biomass areas (section 2). In PISCES-CLG, a sensitivity experiment in which the clearance rate is decreased for chlorophyll concentrations above 0.5 mg Chl m$^{-3}$, the Spearman correlation coefficient is doubled when comparing simulated and observed FFGM concentrations. Note however that this clogging mechanism and its impact on pelagic tunicates growth are largely under-documented, and rely on a few 30-yr old publications (Harbison et al., 1986; Fortier et al., 1994).
However, our modeled variability of the spatial distribution of FFGM was 25 times lower than the observed variability (Table 4). This large variability in observations has already been described in previous compilations of pelagic tunicates observations (Luo et al., 2020, 2022). Numerous aspects may contribute to the high variability of observations compared to models: scarcity of the observations, design of the sampling strategy (Hjøllo et al., 2021), biases in the sampling and enumeration methods (Frank, 1988; Mack et al., 2012), use of species- and location-dependent conversion factors (Arhonditsis
and Brett, 2004), differing definitions of the compared groups or communities and the scale of investigation (local measurements are compared to average 5x5°estimates). Indeed, zooplankton patchiness increases with organism size (Buitenhuis et al., 2013). Physical (mesoscale and submesoscale processes) and biological (diel vertical migrations, predator avoidance, food patches, mate search) processes combine to drive zooplankton patchiness (Folt and Burns, 1999). Although the introduction of a macrozooplankton compartment (namely cnidarian jellyfish) has been shown to increase patchiness in a recent modeling
study (Wright et al., 2021), the spatial resolution (≈2 degrees) of our model setup, and the lack of key biological processes (e.g., complex life cycle) in our model likely preclude the representation of such patchiness. Another source of uncertainty lies in the use of a taxon-specific carbon conversion factor to convert thaliacean abundance data to biomass data. While this approach is appropriate for many protists, thaliacean biomasses estimates based on this method are highly uncertain because these organisms can vary in length by more than one order of magnitude (Iguchi and Ikeda, 2004). In particular, most of the





time, when a net returns hundreds of salps, these salps are relatively young blastozooids (i.e., on the small end of the size
range). Thus estimating biomass from abundance may lead to an overestimation of the true biomass variability. This supports
the need to move towards a systematic reporting of biomass (or at least biovolumes) during zooplankton surveys.

Also, the data temporal resolution is insufficient to analyse seasonal patterns: only 7% of the grid points in the AtlantECO
climatology are derived from data covering at least 6 distinct months. Yet, our standard PISCES-FFGM simulation shows an
approximate one-month lead in the seasonal biomass peak of FFGM compared to GM, this lag being consistent at the global
scale to that of the food of the two groups (Figure A8). This suggests that the filter-feeding mode of FFGM may have an
impact on the temporal dynamics of the FFGM-driven POC flux. However, it is difficult to give a high confidence level to this
statement because the spatial distributions between the lags of the organisms and their food are very patchy and the temporal
variability of the prey does not correspond to that of the corresponding groups when focusing on specific regions (Figure A8).
This claim supports the need to improve the temporal monitoring of FFGM populations in order to understand their seasonality
and thus characterize the seasonal variations of FFGM impacts on carbon fluxes.

## 4.3  Model limitations in representing FFGM

### 4.3.1  Boom-and-burst dynamics

Pelagic tunicates exhibit pullulation-extinction population dynamics, i.e. the alternation between rapid growth phases and
massive mortality events. As a consequence, patchiness is particularly strong for gelatinous zooplankton (Graham et al., 2001;
Purcell, 2009; Lilley et al., 2011; Lucas et al., 2014). However, this dynamics is clearly not simulated by PISCES-FFGM.
This result was expected as biogeochemical models are known to struggle to reproduce the observed spatial variability in the
abundance of different groups of meso- and macro-zooplanktonic organisms (Wright et al., 2021). From a biogeochemical
perspective, the impacts of FFGM on ecosystem structure and carbon export are therefore "smoothed" in time and space when
simulated by PISCES-FFGM. Still, the results obtained provide a first assessment of the annual impacts of FFGM at the global
scale and in large biogeochemical regions (e.g. low productive oligotrophic gyres vs. highly productive upwelling regions).

However, the currently modeled FFGM ability to consume prey over a wide size range is not the only factor likely to
trigger boom-and-bust dynamics. FFGM high clearance rates and complex life cycles with an asexual reproductive phase,
currently not represented in the standard model, are also likely to play a role in such dynamics. In the PISCES-HGR sensitivity
experiment, increasing growth rates of FFGM without adequate modifications of FFGM mortality rates caused the generic
macrozooplankton population to collapse because they were outcompeted by FFGM everywhere except in the mesopelagic
and deep ocean. As expected, and similarly to Luo et al. (2022), the modification of the quadratic mortality in the PISCES-
HGM sensitivity experiment neither improved the fit with the observations, nor triggered any boom-and-burst dynamics. To
further investigate the effect of high growth rates and clearance rates of FFGM, a better understanding of the physiological and
environmental drivers of the FFGM mortality processes triggering the end of their swarms seems essential, as their causes are
multiple and too poorly documented to be currently modeled (Pitt et al., 2014).



Also, life cycles are currently not represented in the model though it could significantly affect the temporal dynamics of a biogeochemical-model (Clerc et al., 2021). Most FFGM have a complex life-cycle, with an alternation between a sexual and asexual phase that could be a major driver of their population dynamics (Henschke et al., 2016). A single-species observation based study on *Thalia democratica* in South-East Australia suggested that life history characteristics such as asexual reproduction and growth are associated with inter-annual variations in abundance and thus may be major factors determining population dynamics, in particular the magnitude of swarms (Henschke et al., 2014). Inclusion of such life cycle traits in a single-species model of *Salpa thompsoni* in the Southern Ocean helped understand the seasonal and interannual variability of salp abundance (Henschke et al., 2018). These studies are focused on one species and one region, and the inclusion of their life cycle in a global model in which FFGM constitute a single compartment would require a multispecies large scale evaluation of the FFGM life cycle role in the temporal dynamics of the swarming process.

### 4.3.2   Carcasses and fecal pellets transfer efficiency

One of the greatest sources of uncertainty about the export of carbon from FFGM to the deep ocean is the transfer efficiency (see Table 6), which depends primarily on remineralization rates and sinking speeds. This raises questions about the processes that could affect the fate of carcasses and fecal pellets (CAFP) as they sink. At a given temperature, our simple FFGM representation includes constant remineralization of CAFP and consumption through filter feeding by GM (Eq. A14 and A15). The induced losses are very low compared to FFGM's CAFP production rates (<5%). However, predation by scavengers could significantly affect CAFP during their fall (Dunlop et al., 2018; Scheer et al., 2022). Benthic consumption by scavengers is well documented for jellyfish carcasses (Sweetman et al., 2014; Henschke et al., 2013), but their fate in the vertical column is largely unknown. Also, parasitism by hyperiid amphipods is likely to affect FFGM carcasses production and degradation, and thus affects deep carbon export by FFGM (Lavaniegos and Ohman, 1998; Phleger et al., 2000; Hereu et al., 2020). Lastly, most measured sinking speed values are based on small (a few meters) sinking column experimental setup and thus do not account for any degradation process (Lebrato et al., 2013). Thus, by combining particularly high velocities with a partial representation of the degradation processes, we mechanistically obtain a particularly high transfer efficiency of FFGM particles. Our estimate of the impact of FFGM on the deep carbon cycle should therefore be interpreted as an upper bound, and a better understanding of FFGM carcasses and fecal pellets fate is needed to properly estimate their impacts on the deep ocean.

## 5   Conclusions

We explicitly represented large pelagic tunicates in the global marine biogeochemistry model PISCES and evaluated the simulated distribution of FFGM by compiling available observations into a FFGM biomass climatology using a taxon-resolving biomass-abundance conversion. Representation of FFGM in a marine biogeochemical model has a small impact on total detritus production in the first 100 m, with 6% of this production due to FFGM. Due to their high sinking speeds, almost all of the organic matter produced by FFGM is transferred to the deep ocean. Therefore, FFGM carcasses and fecal pellets dominate the export of organic matter in the deep ocean (e.g. 70% at 5000 m). The spatial distribution of FFGM-driven export differs





from that of the other macrozooplankton group, GM, which also contributes significantly to export at depth (25% at 5000
m). Indeed, due to their filter-feeding mode of predation, access to preys of variable size allows FFGM to better exploit low
productivity environments than GM, especially in subtropical oligotrophic gyres, where FFGM are twice as abundant as GM
and thus contribute 5 times more to POC export at 1000 m.

A more detailed inclusion of the processes involved in the boom-and-burst dynamics of FFGM (e.g. life cycle, clogging,
high clearance rates) will be necessary to better understand the spatial and temporal variability of their impacts on carbon
export and ecosystem structure. Still, a promising perspective would be to run our PISCES-FFGM model forced by climate
projections. Such a simulation would allow analysis of annual global and large scale regional trends in the impact of FFGM
on marine biogeochemistry. In particular, as climate change could favor small phytoplankton (Peter and Sommer, 2013), we
could expect an amplification of the spatial pattern we currently described, with FFGM even more favored in low productive
regions.

*Code and data availability.*   This section needs to be completed. All raw and gridded data sets will be made publicly available in open access
within the framework of the European H2020 project AtlantECO (grant agreement no 862923). Preliminary DOIs can be made available to
the reviewers upon request. All model outputs necessary to reproduce the results in this manuscript will be made publicly available.

**Appendix A:  Text, figures and tables**

**Appendix content**

1. Text A1 to A3

2. Figures A1 to A8

3. Table A1

**Text A1.**

When including Aus-CPR and SO-CPR data, the resulting point biomass measurements ranged between 0.0 mg C m$^{-3}$ and
19'451 mg C m$^{-3}$, with and average of 0.63 ± 48 mg C m$^{-3}$. However, this range is largely zero-inflated (94.6% of the
observations corresponded to a biomass of 0.0 mg C m$^{-3}$) due to the high relative contribution of both CPR surveys whose data
only comprised 1.1% of non null values. Such strong zero inflation can be attributed to sampling artifacts due to the specificities
of the CPR and thus very likely do not reflect reals absences (Richardson et al., 2006). Indeed, the CPR continuously collects
plankton at standard depth of 7 m and at a speed of nearly 0.2 m s$^{-1}$, as seawater flows in through a square aperture of 1.61
cm$^2$, which is too narrow to adequately sample large gelatinous macrozooplanton such as salps and doliolids, especially in
the Southern Ocean (Pinkerton et al., 2020). Consequently, we decided to remove the observations from the AusCPR and the
SO-CPR from our final validation data set.





**Text A2.**

**Nutrients**

Map a. (resp. b.) in Fig. A2 presents the observed (resp. simulated) surface concentrations of nitrates. The model performs
particularly well for surface nitrates, with absolute values and simulated spatial patterns very consistent with observations
(r=0.83). The model performance is very similar for phosphates (r=0.83).

**Chlorophyll**

The modeled annual chlorophyll distribution is compared to OC-CCI satellite observations in Fig. A2 c. and d. The correspon-
dence between the observed and simulated surface chlorophyll is rather satisfactory (r= 0.59). The average value is similar
(0.37 vs 0.42 mg Chl m$^{-3}$) and the spatial structure is respected overall. The overall variability is of the same order of mag-
nitude in the model and the observations (standard deviation of 0.32 mg Chl m$^{-3}$ for the model and 0.64 mg Chl m$^{-3}$ for the
observations). However, there are some differences. At high latitudes, particularly in the Southern Ocean, the model tends to
overestimate chlorophyll compared to the satellite product. However, satellite chlorophyll may be underestimated by a factor
of about 2 to 2.5 by the algorithms deducing chlorophyll concentrations from reflectance as discussed in Aumont et al. (2015).

**Mesozooplankton**

Mesozooplankton annual distribution on the top 300 m is compared to the MAREDAT product in Fig. A2 e. and f. The model
performs quite well (r=0.45) and fits the observed spatial patterns, and the distribution of high vs. low concentration regions.
However, it tends to overestimate the low concentrations and underestimate the high concentrations. Indeed, mesozooplankton
variability is slightly reduced in the model (standard deviation of 0.34 vs 0.59 mmol C m$^{-3}$ in the observation).

**Text A3.**

**Macrozooplankton dynamics:**

$G_X$, the ingested matter, is depending on food availability to $X$. We distinguish two predation behaviours: concentration-
dependent grazing and flux feeding.
Concentration-dependent grazing is based on a Michaelis-Menten parameterization with no switching and a threshold (Gen-
tleman et al., 2003). The equation describing the grazing rate of $X$ on prey $I$, $g^X(I)$, is derived as:

$$F^X = \sum_J p_J^X \max\left(0, J - J_{\text{thresh}}^X\right) \tag{A1}$$

$$F_{\text{lim}}^X = \max\left(0, F^X - \min\left(0.5F, F_{\text{thresh}}^X\right)\right) \tag{A2}$$





$$g^X(I) = g_m^X \frac{F_{\text{lim}}}{F} \frac{p_I^X \max\left(0, I - I_{\text{thresh}}^X\right)}{K_G^X + \sum_J p_J^X J} \tag{A3}$$

where $F^X$ is the available food to $X$, $g_m^X$ is the maximal grazing by $X$ rate, $F_{\text{thresh}}^X$ is the feeding threshold for $X$, $I_{\text{thresh}}^X$ is the group $I$ threshold for $X$, $K_G^X$ is the half saturation constant for grazing by $X$, $p_I^X$ is the $X$ preference for group $I$.

Flux-feeding accounts for particles traps deployed by some zooplankton species (Jackson, 1993). It is derived as a particles flux depending term, an thus depends on the product of the concentration by the sinking speed:

$$\text{ff}^X(I) = \text{ff}_m^X w_I I \tag{A4}$$

where $\text{ff}^H(I)$ is the flux-feeding rate of prey $X$ on particle $I$, $\text{ff}^H(I)$ is the maximal flux-feeding rate of prey $X$ on particle $I$, $w_I$ is the vertical sinking velocity of $I$ particles.

For GM:

$$G_{GM}^g = g^{GM}(P) + g^{GM}(D) + g^{GM}(\text{sPOC}) + g^{GM}(\text{bPOC}) + g^{GM}(Z) + g^{GM}(M) \tag{A5}$$

$$G_{GM}^{\text{maxff}} = \text{ff}^{GM}(\text{bPOC}) + \text{ff}^{GM}(\text{sPOC}) + \text{ff}^{GM}(Ca_{GM}) + \text{ff}^{GM}(Fp_{GM}) + \text{ff}^{GM}(Ca_{FFGM}) + \text{ff}^{GM}(Fp_{FFGM}) \tag{A6}$$

$$E_{GM}^{\text{ff}} = \frac{G_{GM}^{\text{maxff}}}{G_{GM}^g + G_{GM}^{\text{maxff}}} \tag{A7}$$

$$G_{GM}^{\text{ff}} = G_{GM}^{\text{maxff}} E_{GM}^{\text{ff}} \tag{A8}$$

$$G^{GM} = G_{GM}^{\text{ff}} + G_{GM}^g \tag{A9}$$

$$p_M^{GM} >> p_D^{GM} = p_Z^{GM} \tag{A10}$$

with $E_{GM}^{\text{ff}}$ the proportion of filter-feeders, $G_{GM}^{\text{maxff}}$ the potential ingestion by flux feeding, $G_{GM}^{\text{ff}}$ the actual ingestion by flux
feeding , $G_{GM}^g$ the ingestion by concentration dependent grazing and $p_Y^X$ the $X$ preference for group $Y$

For FFGM:

$$G_{FFGM} = g^{FFGM}(P) + g^{FFGM}(D) + g^{FFGM}(\text{POC}) + g^{FFGM}(\text{GOC}) + g^{FFGM}(Z) + g^{FFGM}(M) \tag{A11}$$





$$p_D^{FFGM} = p_N^{FFGM} = p_Z^{FFGM} \tag{A12}$$

For the PISCES-CLG experiment (with FFGM clogging) run, the ingested matter by FFGM $G_{FFGM}^{CLG}$ is:


$$G_{FFGM}^{CLG} = G_{FFGM} \times F_C(Chl) \tag{A13}$$

where $F_C(Chl)$ is the clogging function presented in Eq. 2 of the paper.

**Carcasses dynamics:**

Carcasses production by organisms $X$ (=$FFGM$ or =$GM$) comes from non predatory quadratic and linear $X$ mortalities.
Loss terms include a temperature dependent term representing remineralization by saprophagous organisms and flux-feeding

by GM. Flux feeding includes two terms : the ingested food by GM which is temperature dependent and the non ingested
matter fractionated by flux feeding process (Dilling and Alldredge, 2000), which is assumed to be equal to the ingested portion
except the temperature dependency.

$$
\begin{aligned}
\frac{\partial Ca_X}{\partial t} + w_{Ca_X} \frac{\partial Ca_X}{\partial z} \quad &= m_c^X f_X(T) \left(1 - \Delta(O_2)\right) X^2 \\
&+ r^X f_X(T) \left( \frac{X}{K_m + X} + 3\Delta(O_2) \right) X \\
&- E_{GM}^{\mathrm{ff}} \mathrm{ff}^{GM}(Ca_X)(1 - \Delta(O_2)) f_{GM}(T) GM \\
&- E_{GM}^{\mathrm{ff}} \mathrm{ff}^{GM}(Ca_X) GM \\
&- \alpha f_\alpha(T) Ca_X
\end{aligned}
\tag{A14}
$$


Where $\alpha$ is the remineralization rate.

**Fecal pellets dynamics:**

Fecal pellets production by organisms $X$ (=$FFGM$ or =$GM$) comes from non assimilated food. Loss terms, similarly to
carcasses, include a temperature dependent remineralization term and a flux-feeding by GM term.

$$
\begin{aligned}
\frac{\partial Fp_X}{\partial t} + w_{Fp_X} \frac{\partial Fp_X}{\partial z} \quad &= a^X I_X^g \left(1 - \Delta(O_2)\right) f_X(T) X \\
&- E_{GM}^{\mathrm{ff}} \mathrm{ff}^{GM}(Fp_X)(1 - \Delta(O_2)) f_{GM}(T) GM \\
&- E_{GM}^{\mathrm{ff}} \mathrm{ff}^{GM}(Fp_X) GM \\
&- \alpha f_\alpha(T) Fp_X
\end{aligned}
\tag{A15}
$$


Where $a^X$ is the $X$ assimilation rate.



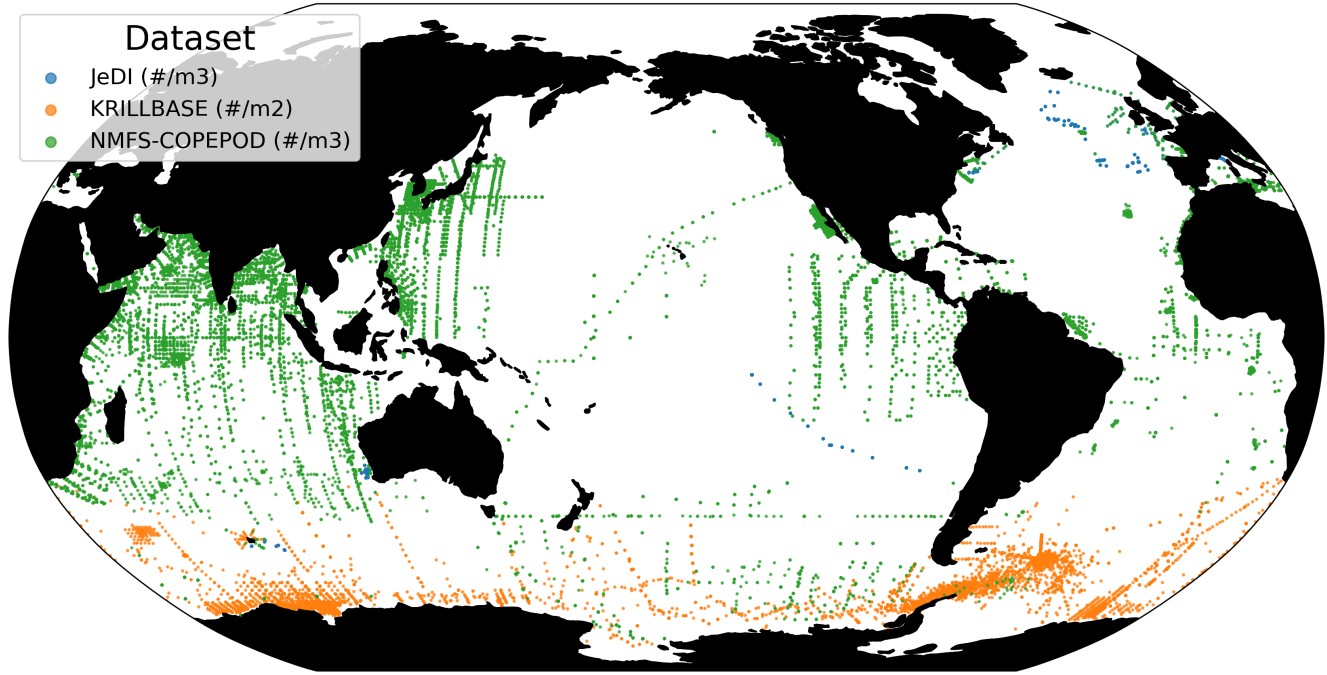

**Figure A1.** Map of the FFGM observations in the AtlanECO product. Colors indicate the original dataset.





**Figure A2. Comparison between modeled and observed annual average surface nitrates (a. and b.), surface chlorophyll (c. and d.) and mesozooplankton biomass integrated over the top 300 m (e. and f.)** The mesozooplankton field (mmol m$^{-3}$, vertically integrated between 0 and 300 m) from MARine Ecosystem DATa (MAREDAT) (Moriarty and O'Brien, 2013) is used to evaluate our modeled total mesozooplankton biomass distribution. The $PO_4^{3-}$ and $NO_3^-$ surface fields from the World Ocean Atlas (Garcia et al., 2019) are used to evaluate our modeled nutrient distributions. The long-term multi-sensor time-series OC-CCI (Ocean Colour project of the ESA Climate Change Initiative, Sathyendranath et al. (2019)) of satellite phytoplankton chlorophyll-*a* sea surface concentration converted into mmol m$^{-3}$ is used to evaluate our modeled total chlorophyll distribution.





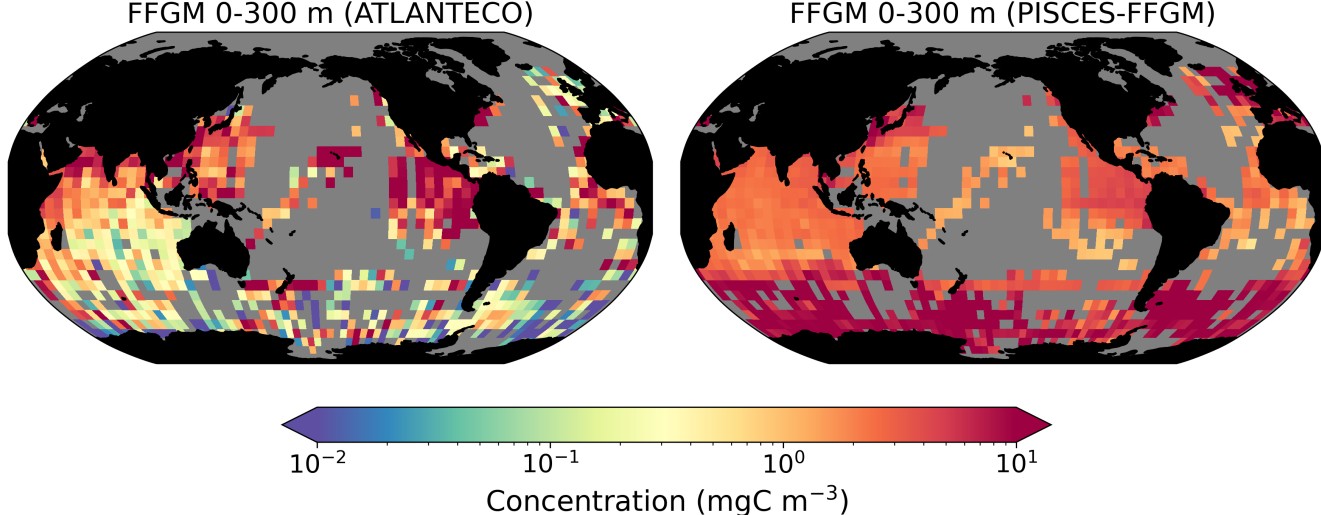

**Figure A3. Comparison between AtlantECO observed and PISCES-HGR modeled FFGM biomasses.** The colobars are in logarithmic scale. a. Annual average of monthly observations of FFGM concentrations Atlanteco on 5 degree resolution grid. b. Annual average of monthly modeled FFGM concentrations by PISCES-HGR on 5 degree grid masked for missing monthly observations.



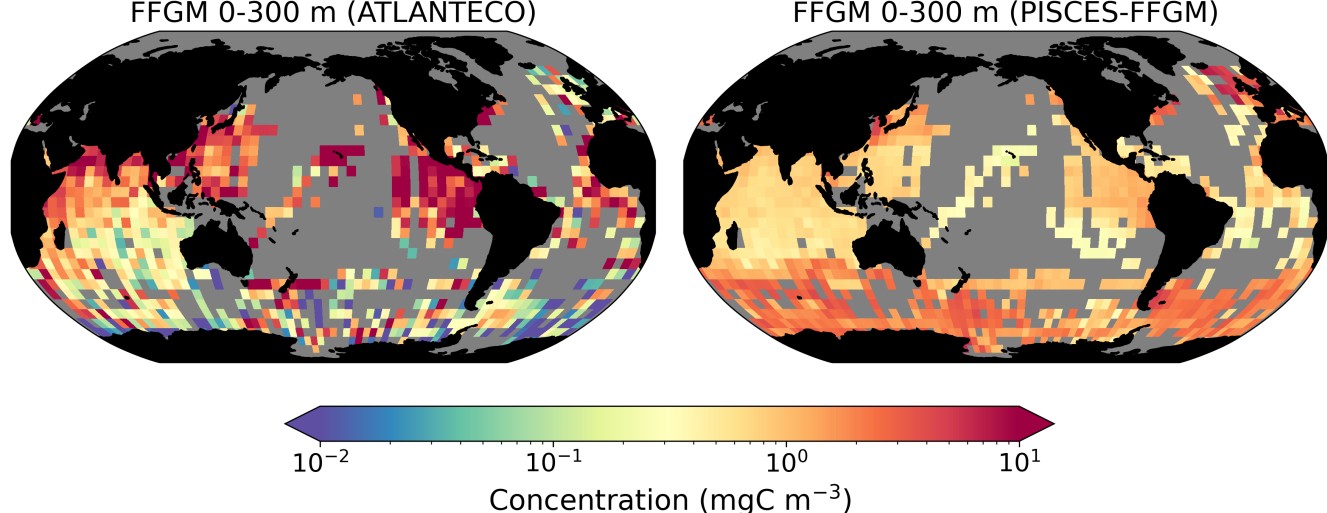

**Figure A4. Comparison between AtlantECO observed and PISCES-HGM modeled FFGM biomasses.** The colobars are in logarithmic scale. a. Annual average of monthly observations of FFGM concentrations Atlanteco on 5 degree resolution grid. b. Annual average of monthly modeled FFGM concentrations by PISCES-HGM on 5 degree grid masked for missing monthly observations.



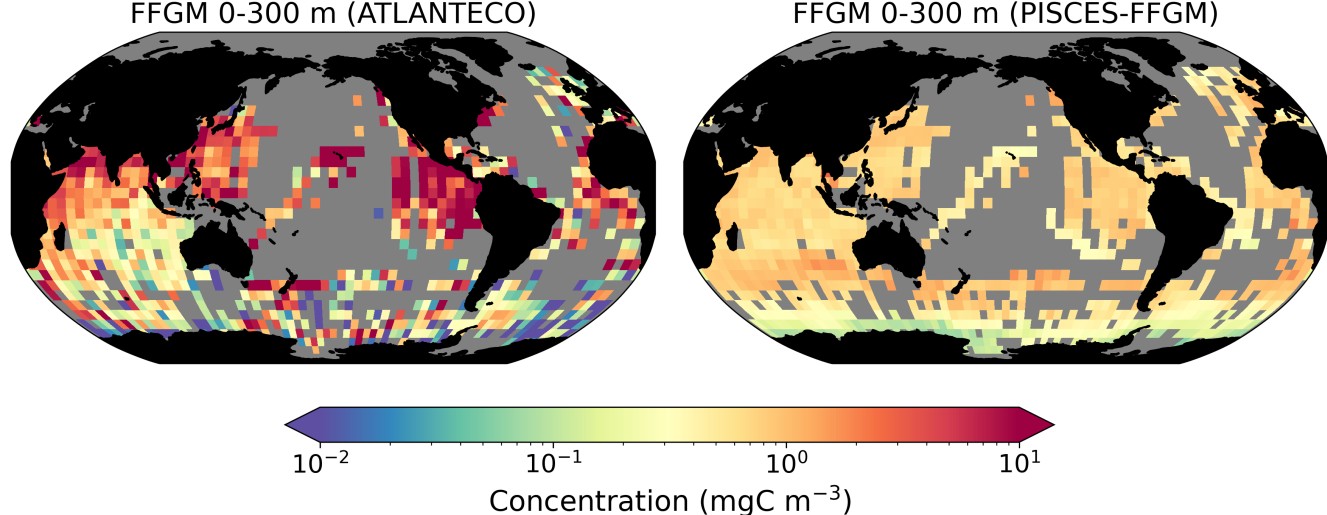

**Figure A5. Comparison between AtlantECO observed and PISCES-CLG modeled FFGM biomasses.** The colobars are in logarithmic scale. a. Annual average of monthly observations of FFGM concentrations Atlanteco on 5 degree resolution grid. b. Annual average of monthly modeled FFGM concentrations by PISCES-CLG on 5 degree grid masked for missing monthly observations.





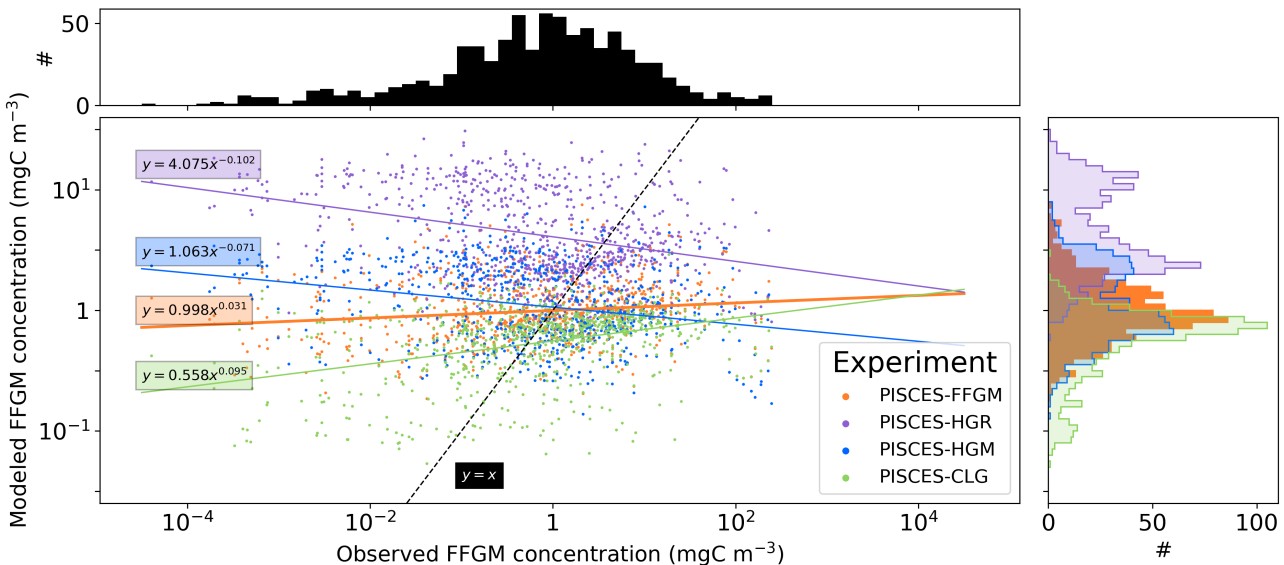

**Figure A6. Observation-model scatter plot and histogram of FFGM observed and modeled biomass values distribution** 72x36 monthly gridded product are used for both modeled and observed FFGM biomasses. Linear regression are applied to each model (plain lines).



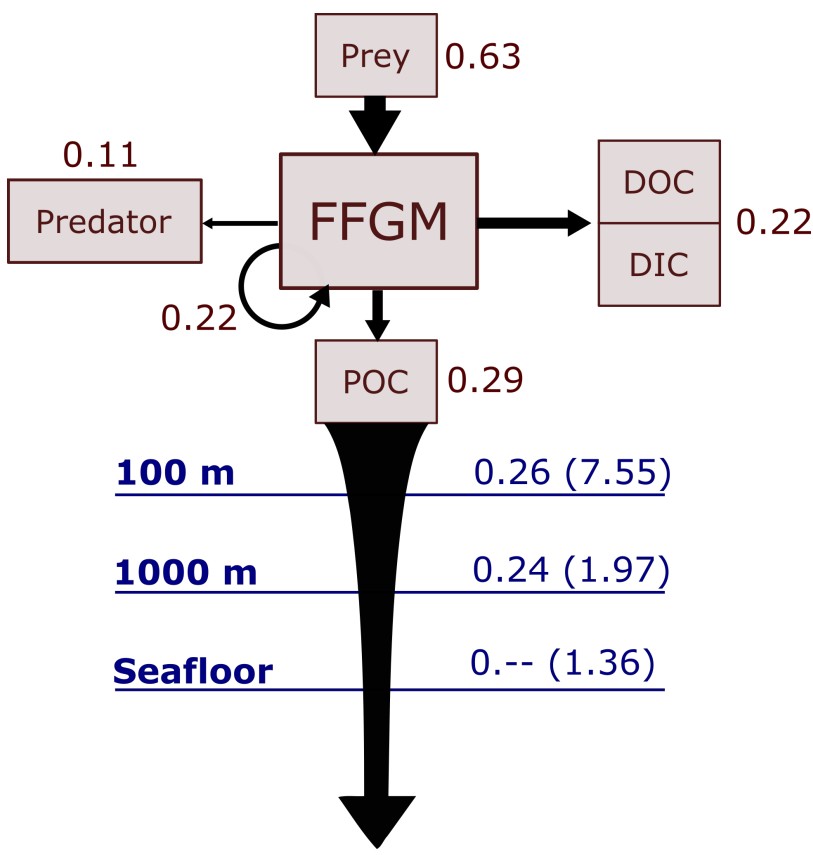

**Figure A7. Schematic representation of carbon fluxes induced by processes related to GM.** Values are in PgC/year. The upper part of the diagram represents the inflows and outflows of GMs integrated globally over the first 100 meters. The inflow is the grazing on the different prey. The arrow going from GM to GM corresponds to the flux related to growth due to assimilated food. The outflows are : i) the remineralization/non-assimilation processes that go into the dissolved organic carbon (DOC) and dissolved inorganic carbon (DIC) ii) the quadratic and linear mortality terms (directly remineralised in PISCES-FFGM because of the lack of explicit representation of upper level predators) and iii) the production of particular organic carbon (POC) via carcasses and fecal pellets. The lower part of the diagram corresponds to the export of POC linked to the fall of carcasses and fecal pellets of GM. The values in blue correspond to the global annual GM-driven POC flux through the corresponding depth, the values in parenthesis representing the total POC flux (related to FFGM, GM, bPOC and sPOC).





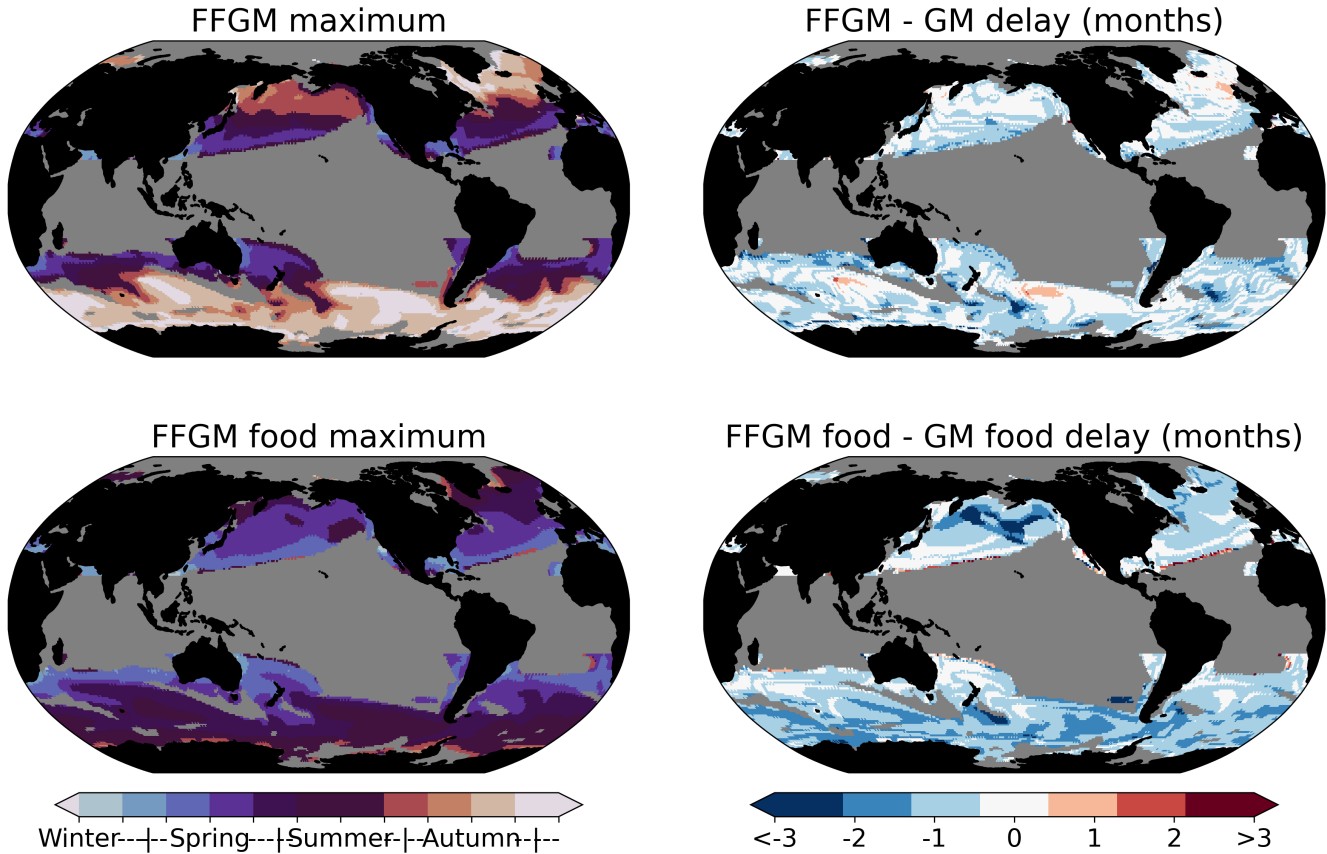

**Figure A8. Spatial distribution of the annual period of maximum macrozooplankton biomasses and maximum food availability** A filter was applied to keep only the areas at more than 20°latitude from the equator and in which the amplitude of annual biomass variation is higher than 20%. The amplitude is calculated as $(2 \times (max - mix)/(min + max))$ with $min$ the minimum annual biomass and $max$ the maximum annual biomass. a. Map of months with maximal FFGM biomasses b. Map of lag (in months) between months of maximal FFGM biomasses and months of maximal FFGM biomasses c. Map of months with maximal FFGM food availability (calculated as the sum of prey weighted by FFGM preferences for each prey) d. Map of lag (in months) between months with maximal FFGM food availability and months with maximal GM food availability.





| Class | Order | Genus | Species | Individual weight (mg C ind$^{-1}$) | Source |
|---|---|---|---|---|---|
| Thaliacea | Doliolida | *Dolioletta* | *gegenbauri* | 0.0192 | (Lucas et al., 2014) |
| Thaliacea | Pryosomatida | *Pryosoma* | *atlanticum* | 22.9036 | (Lucas et al., 2014) |
| Thaliacea | Salpida | *Brooksia* | *rostrata* | 0.0019 | (Lucas et al., 2014) |
| Thaliacea | Salpida | *Cyclosalpa* | *affinis* | 2.8196 | (Lucas et al., 2014) |
| Thaliacea | Salpida | *Cyclosalpa* | *bakeri* | 4.7948 | (Lucas et al., 2014) |
| Thaliacea | Salpida | *Cyclosalpa* | *floridana* | 0.1146 | (Lucas et al., 2014) |
| Thaliacea | Salpida | *Cyclosalpa* | *pinnata* | 3.473 | (Lucas et al., 2014) |
| Thaliacea | Salpida | *Cyclosalpa* | *polae* | 0.5262 | (Lucas et al., 2014) |
| Thaliacea | Salpida | *Iasis* | *zonaria* | 3.9887 | (Lucas et al., 2014) |
| Thaliacea | Salpida | *Ihlea* | *punctata* | 0.1673 | (Lucas et al., 2014) |
| Thaliacea | Salpida | *Pegea* | *bicaudata* | 7.9575 | (Lucas et al., 2014) |
| Thaliacea | Salpida | *Pegea* | *confoederata* | 1.8974 | (Lucas et al., 2014) |
| Thaliacea | Salpida | *Pegea* | *socia* | 1.6717 | (Lucas et al., 2014) |
| Thaliacea | Salpida | *Salpa* | *aspera* | 2.9474 | (Lucas et al., 2014) |
| Thaliacea | Salpida | *Salpa* | *cylindrica* | 0.56 | (Lucas et al., 2014) |
| Thaliacea | Salpida | *Salpa* | *fusiformis* | 1.33 | (Lucas et al., 2014) |
| Thaliacea | Salpida | *Salpa* | *maxima* | 3.2305 | (Lucas et al., 2014) |
| Thaliacea | Salpida | *Thalia* | *democratica* | 0.042 | (Lucas et al., 2014) |
| Thaliacea | Salpida | *Thetys* | *vagina* | 0.404 | (Lucas et al., 2014) |
| Thaliacea | Salpida | *Salpa* | *thompsoni* | 10.57 | (Kiørboe, 2013) |

**Table A1. Table of individual weights used for abundance to biomass conversions** For *Salpa thompsoni*, we computed the mean of the corresponding mass measurements of individual zooplankters in table A1 of Kiørboe (2013). For all the other species, we used values from Appendix S4 from Lucas et al. (2014)



*Competing interests.* The authors declare no competing interests

*Acknowledgements.* We are very grateful to Lars Stemmann, Olivier Maury, Jean-Christophe Poggiale and Fabien Lombard for insightful comments during the development of this manuscript and to Christian Ethé and Olivier Torres for setting up the model configuration.

This project used the HPC resources of TGCC under the allocation NUMBER (project gen0040) provided by GENCI (Grand Equipement National de Calcul Intensif). This study benefited from the ESPRI (Ensemble de Services Pour la Recherche l'IPSL) computing and data center (https://mesocentre.ipsl.fr) which is supported by CNRS, Sorbonne Université, Ecole Polytechnique, and CNES and through national and international grants.

   This study has received funding from the Agence Nationale de la Recherche grant agreement ANR-17-CE32-0008 (CIGOEF).

MV and FB aknowledge funding from the European Union's Horizon 2020 research and innovation programme under grant agreement no. 862923. This output reflects only the author's view, and the European Union cannot be held responsible for any use that may be made of the information contained therein.

   LB acknowledges support from the European Union's Horizon 2020 research and innovation COMFORT (grant agreement No 820989), ESM2025 (grant agreement No 101003536) and from the Chaire ENS-Chanel.



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
