# Peer review of "Including filter-feeding gelatinous macrozooplankton in a global marine biogeochemical model: model-data comparison and impact on the ocean carbon cycle"

_EGUsphere, 2022_

## Referee Comment (RC1)

**General Comments**

I think the authors have offered an interesting, thorough, and generally well-presented manuscript. The manuscript adds a valuable contribution to the growing body of research around modelling gelatinous zooplankton, further supporting the importance of gelatinous zooplankton in global carbon export while also highlighting key areas for future research.

**Specific Comments**

Consider using 'Crustacean Macrozooplankton (CM)', instead of 'Generic Macrozooplankton (GM)'.

> As the generic macrozooplankton group is based entirely on crustacean macrozooplankton I think this would aid in the clarity and precision of the manuscript. Generic implies that the group represents organisms across the macrozooplankton i.e. more than just crustaceans.

Line 151

> Include an explanation on why the decision was made to have 'feeding with identical preferences on both phytoplankton groups… as well as on microzooplankton'.

Section 2.3 Sensitivity experiments

> Adding an introductory sentence to this section would help the flow and to guide the reader, i.e. 'Five sensitivity experiments were carried out to…'.

Rainbow colour palette for figures (fig. 3a, fig. 4, fig. 8, fig. A2e,f, fig. A3, fig. A4, fig. A5)

> I strongly recommend changing from a rainbow colour palette for figures, to a "perceptually uniform" palette which provides the same perceived colour change over the same change in value.

Line 286

> Include a brief mention of why 0-300m was chosen for analysis.

Line 324-325

> It is interesting that doubling the complexity of the zooplankton (from 2 to 4 compartments) has minimal effect on chlorophyll. It would be interesting to have a short discussion on why this is and how it compares to other biogeochemical models.

Section 3.1.2

> What is the reasoning behind using 0.5 mg C m$^{-3}$ as the cut off for high/low biomass (Table 4 etc).

Figure 5

> Red/green is not a good choice for distinguishing between the two model runs, as red/green is the most common type of colour-blindness. This should be changed to a colour-blind friendly palette.

Table 6

> Include a description of the difference between the two Luo et al. (2020) columns.

**Technical Corrections**

Line 2

> '…that feed on preys' should be '…that feed on prey'

Line 29

> '…biological carbon cycling through "jelly-falls" events'
> The word 'events' is unnecessary in this sentence and should be removed.

Line 73

‘…the PLANKTOM model’ should be ‘…the PlankTOM model’

Line 323

It feels odd to start a new section with ‘Then’, perhaps change for ‘Next’.

Table 4

Capitalise ‘Comparison’ in column one

Line 380

‘sensitivity experiment’ should be ‘sensitivity experiments’

Figures A3, A4, A5

Labelling is incorrect on the model map for each of these figures. A3 should be PISCES-HGR, A4 should be PISCES-HGM, A5 should be PISCES-CLG.

Figure A8

Panels are not labelled with the a, b, c, d indicated in the figure description.

---

## Author Response (AR2)

*Title: Including filter-feeding gelatinous macrozooplankton in a global marine biogeochemical model: model-data comparison and impact on the ocean carbon cycle*
*Author(s): Corentin Clerc et al.*
*MS No.: egusphere-2022-1282*
*MS type: Research article*

Dear Associate Editor,

We are pleased to submit a revised version of our manuscript entitled "*Including filter-feeding gelatinous macrozooplankton in a global marine biogeochemical model: model-data comparison and impact on the ocean carbon cycle*" (ID *egusphere-2022-1282*) as well as a point-by-point reply to the comments, and a marked-up manuscript version showing the changes made. We thank reviewer #1 and #2 as well as the Associate Editor for their supportive and relevant comments which have improved the manuscript.

We believe that the changes made have improved the manuscript and hope that it is now suitable for publication in *Biogeosciences.*

Yours sincerely,
Corentin CLERC

On behalf of all co-authors

**Point-by-point reply to reviewers' comments**

The authors would like to thank the two reviewers for their valuable and useful comments. The authors considered all suggestions and addressed the raised issues, which we believed increased the clarity of the revised manuscript. Below are given point-by-point answers to the comments.

Reviewers comments are in bold, responses in normal font. Changes to the manuscript are in italics. Line numbers mentioned below refer to the revised manuscript.

**REVIEWER #1**

**I think the authors have offered an interesting, thorough, and generally well-presented manuscript. The manuscript adds a valuable contribution to the growing body of research around modelling gelatinous zooplankton, further supporting the importance of gelatinous zooplankton in global carbon export while also highlighting key areas for future research.**

We would like to thank Reviewer #1 for her very supportive comments on our manuscript.

**SPECIFIC COMMENTS**

**Consider using 'Crustacean Macrozooplankton (CM)', instead of 'Generic Macrozooplankton (GM)'. As the generic macrozooplankton group is based entirely on crustacean macrozooplankton I think this would aid in the clarity and precision of the manuscript. Generic implies that the group represents organisms across the macrozooplankton i.e. more than just crustaceans.**

We agree with the reviewer that the definition of the generic macrozooplankton compartment (L136) was not precise enough, which may have lead to some confusion. Actually, our 'generic macrozooplankton group' intends to represent macrozooplankton that are not part of the Tunicata ("non-tunicate macrozooplankton"), which includes a lot of crustaceans but not only (e.g., pteropods). Since this compartment was parameterized using an allometric scaling relationship rather than group-specific parameterizations, we think that the group is generic enough to represent also non-crustacean types of macrozooplankton, such as pteropods. For this reason, the generic macrozooplankton notation is more appropriate, and the corresponding explanatory text in the Methods has been corrected as follows:

L136: *"GM, namely generic macrozooplankton, is intended to represent non-tunicate macrozooplankton, such as euphausids, pteropods or large copepods.»* instead of *"GM, namely generic macrozooplankton, is intended to represent crustacean macrozooplankton, such as euphausids or large copepods.»*

Also, L137, *"The parametrization is similar to that of mesozooplankton (Eq. 28 to 31 in Aumont et al. 2015)"* have been replaced by *"The parametrization is similar to that of mesozooplankton (Eq. 28 to 31 in Aumont et al. 2015) and parameter values have been derived using allometric scaling relationships (see section 2.2.1)."*

**Line 151:**
**Include an explanation on why the decision was made to have 'feeding with identical preferences on both phytoplankton groups... as well as on microzooplankton'.**

We assumed non-selective predation when modelling filtration in FFGM, meaning that the organisms are able to ingest all filtered prey. This lack of food selectivity is a known characteristic of filter-feeders (Pauli. et al., 2021b). Recent studies suggest that the efficiency of prey capture by filter-feeding gelatinous organisms could depend on the prey size but also on their taxonomy (Sutherland et al., 2022). Yet, while "future work should bring more quantitative studies of grazing rates and selectivity under in situ conditions, expand the number of pelagic tunicate species that have been studied" (Sutherland et al., 2022), quantitative estimates remain insufficient to properly assess this phenomenon, therefore we believe that non-selective predation remains the best option in the current state of knowledge.

To address the present comment, we inserted the following sentence at line 150 : *"Although there is some recent evidence for selective feeding behavior in pelagic tunicates (Sutherland et al. 2022), the lack of quantitative study led to the simpler representation of FFGM as non-selective feeders (Pakhomov et al. 2002, Vargas et al. 2004, von Harbou et al. 2011). Therefore, we assume [...]"*

New references :

von Harbou, L., Dubischar, C. D., Pakhomov, E. A., Hunt, B. P., Hagen, W., and Bathmann, U. V.: Salps in the Lazarev Sea, Southern Ocean: I. Feeding dynamics, Marine Biology, 158, 2009–2026, 2011.

Vargas, C. A. and Madin, L. P.: Zooplankton feeding ecology: clearance and ingestion rates of the salps Thalia democratica, Cyclosalpa affinis and Salpa cylindrica on naturally occurring particles in the Mid-Atlantic Bight, Journal of plankton research, 26, 827–833, 2004.

**Section 2.3: Sensitivity experiments**
**Adding an introductory sentence to this section would help the flow and to guide the reader, i.e. 'Five sensitivity experiments were carried out to...'.**

In accordance with the reviewer's comment, the following introductory sentence has been added at line 194: *'Five sensitivity experiments were carried out to assess the sensitivity of the model to the chosen parameterizations."*

**Rainbow colour palette for figures (fig. 3a, fig. 4, fig. 8, fig. A2e,f, fig. A3, fig. A4, fig. A5)**
**I strongly recommend changing from a rainbow colour palette for figures, to a "perceptually uniform" palette which provides the same perceived colour change over the same change in value.**

In accordance with the suggestion by the reviewer, the colour palette has been changed to a "perceptually uniform" palette in fig. 3a, fig. 4, fig. 8, fig. A2 e,f, fig. A3, fig. A4 and fig. A5. We use the "cividis" colour palette from the matplotlib python package.

**Line 296:**
**Include a brief mention of why 0-300m was chosen for analysis.**

The 0-300 m depth range was chosen for our analyses to ensure that we captured most of the organisms present in the epipelagic zone, which can be as deep as 250-300 m in the tropical gyres. This choice was applied for both macro- and mesozooplankton but there was an error in the manuscript indicating a 0-100 m depth range for macrozooplankton (L289). This typo has been corrected.

We now justify this choice here with the AtlantECO dataset (see section 2.4.1) for FFGM species.

First, if we look at the distribution of the maximal sampling depth (see figure below), there is a clear cut off in the distribution at around 300 m, samples being sporadically distributed for deeper maximal depths. The spatial resolution of data below 300m is therefore very coarse. Furthermore, the depth distribution shown below indicates that using 100 m as a maximal depth would exclude a significant portion of the dataset.

Second, the available data suggest that most of the biomass is present in the first 300 meters. Indeed, when analyzing the raw AtlantECO dataset (i-e non gridded), the mean biomass for FFGM is 4.93 mg C m$^{-3}$ and the median 0.31 mg C m$^{-3}$ for the data with a maximum sampling depth shallower than 300 m. These values drop to 0.04 mg C m$^{-3}$ for the mean and 0.0 mg C m$^{-3}$ for the median when focusing on data with a minimum sampling depth deeper than 300 m.

Last, when focusing on the median by maximum depth bins (red points on the figure below), we show that the median is null from 300m to about 1000 m. Deeper values (>1000m) are very uncertain due to the few data points available as shown on the depth histogram.

[Figure]

Figure A2. Depth-Biomass scatter plot and histogram of FFGM observed biomass and maximal depth of the samples AtlantECO dataset before excluding deep samples (Section 2.4.1). Blue points are samples. The red dots represent the median biomass per depth bin.

Thus, we added the following at L289 and L296: "[*vertically integrated between 0 and 300 m*] to ensure that most of the organisms present in the epipelagic zone are included"

We also replaced L264 *"Then, we only retained observations from the upper 300 m to exclude deep water samples and focus on zooplankton communities that inhabit the euphotic layer"* by *"Then, we only retained observations from the upper 300 m to exclude deep water samples and focus on zooplankton communities that inhabit the epipelagic layer because measured biomasses and sample numbers are low below 300 m (see. Fig. A2)"*

And added the figure above as figure A2 in the appendix (which now contains 10 figures).

**Line 324-325:**
**It is interesting that doubling the complexity of the zooplankton (from 2 to 4 compartments) has minimal effect on chlorophyll. It would be interesting to have a short discussion on why this is and how it compares to other biogeochemical models.**

Chlorophyll is relatively little impacted by the addition of higher trophic levels (in PISCES-FFGM compared to PISCES-v2, the version published in Aumont et al. 2015). Indeed, macrozooplankton represent only 10% of the biomass of all organisms, and they consume less than 5% of the total primary production. Therefore, adding more macrozooplankton groups has a small effect on primary production in our model because grazing rates on phytoplankton are very low (< 10%), partly because GM eats little phytoplankton and then because metabolic rates are low. These low differences in chlorophyll fields also suggest that the mesozooplankton quadratic mortality used in PISCES-v2 is quite relevant to implicitly account for the effect of predation pressure by upper trophic levels on primary productivity.

Note that our study focuses on the role of FFGM, and thus our analysis is mostly based on comparing PISCES-GM with PISCES-FFGM. The modest role of adding FFGM on lower trophic levels (phytoplankton and microzooplankton) is by the way mentioned and explained in Section 4.1.2.

**Section 3.1.2:**
**What is the reasoning behind using 0.5 mg C m$^{-3}$ as the cut off for high/low biomass (Table 4 etc).**

The cut-off value corresponds to the rounded median value of macrozooplankton observations from MAREDAT (0.52 mg C m$^{-3}$), see Table 4. Regions where biomass (simulated or observed) is higher (resp. lower) than the cut-off are defined as high (resp. low) biomass regions.

The following sentence has been added at the end of the legend of Table 4 :

*"The cut-off value of 0.5 mg C m$^{-3}$, used for defining high and low biomass regions, corresponds to the rounded median value of the macrozooplankton observations from MAREDAT (see section 2.4.2)."*

**Figure 5:**
**Red/green is not a good choice for distinguishing between the two model runs, as red/green is the most common type of colour-blindness. This should be changed to a colour-blind friendly palette.**

We switched our previous color palette to a colour-blind friendly one in figure 5.

**Table 6:**
**Include a description of the difference between the two Luo et al. (2020) columns.**

The following sentence has been included in the legend of Table 6: *"There are two columns for Luo et al. (2020) as the authors tested two parameterizations of carcass and fecal pellet sinking speeds: 1000 m/d for carcasses and 650 m/d for fecal pellets (third column) or 800 m/d for carcasses and 100 m/d for fecal pellets (fourth column). »*

**TECHNICAL CORRECTIONS**

All technical corrections listed below have been applied to the revised version of the manuscript.

**Line 2**
**'...that feed on preys' should be '...that feed on prey'**

**Line 29**

'...biological carbon cycling through "jelly-falls" events'
The word 'events' is unnecessary in this sentence and should be removed.

**Line 73**

'...the PLANKTOM model' should be '...the PlankTOM model'

**Line 323**

It feels odd to start a new section with 'Then', perhaps change for 'Next'.

**Table 4**

Capitalise 'Comparison' in column one

**Line 391**

'sensitivity experiment' should be 'sensitivity experiments'

**Figures A3, A4, A5**

Labelling is incorrect on the model map for each of these figures. A3 should be PISCES- HGR, A4 should be PISCES-HGM, A5 should be PISCES-CLG.

**Figure A8**

Panels are not labelled with the a, b, c, d indicated in the figure description.

**REVIEWER #2**

**This is an excellent paper tackling to evaluate the impact of large filter-feeding gelatinous zooplankton (i.e. pelagic tunicates, except for appendicularians) on the marine geochemical cycle on a global scale. Although there will be some uncertainties about the calculated figures due to the paucity of data on the distribution, I believe it is worthwhile to publish them as a basis for raising the issue and for discussion. The manuscript is well prepared and I have no critical comments for the publication except for minor points as shown below.**

We would like to thank Reviewer #2 for her/his very supportive comments on our manuscript.

**1. Some large salp species commonly show a large extent of diel vertical migration. So authors need to mention how they compile the data sets taken at different timing of the day. Also, the effect of the diel vertical migration on carbon transportation through the mesopelagic migrant pump (Boyd et al. 2019) could be added to the discussion.**

The data retained for our analyses lie within the first 300 meters of the ocean's water column. This depth range was indeed chosen to capture the majority of diel vertical migration (DVM) behavior of the target PFT, and to reflect the observational biomass data. Indeed, most of the observed Thaliacea biomass is located in the upper 300 m of the ocean. When analyzing the raw AtlantECO dataset (i-e non gridded), the mean biomass for FFGM is 4.93 mg C m$^{-3}$ and the median 0.31 mg C m$^{-3}$ for the data with a maximum sampling depth shallower than 300 m. These metrics drop to 0.04 mg C m$^{-3}$ for the mean and 0.0 mg C m$^{-3}$ for the median when focusing at the data with a minimum sampling depth deeper than 300 m.

We agree that tunicate biomass estimates are likely to be underestimated since some salp species can migrate several hundred meters per day (e.g. 600 m for *Salpa fusiformis* (Pascual et al. 2017), 300 m for *Salpa thompsonii* (Henschke et al. 2021)). Accounting for this bias as a function of sampling time would require 1) having a clear quantification of migration depth intervals for each tunicate species in the observation dataset, and 2) having an idea of the spatial heterogeneity of the migrations for a given species. However, these data are only available for a small number of species and are based on a few local studies that do not show the spatial variability of migration depths, and that are thus not representative of processes at global monthly scales. For all these reasons, we opted for the simplest approach, namely to average the data according to the month of collection independently of time of the day. We added to the manuscript a mention of this potential bias in the observational data by adding the following sentence (L273): *"Although some pelagic tunicate species show a large extent of diel vertical migration (Pascual et al. 2017, Henschke et al. 2021), the present observational data were averaged per months regardless of sampling time, due to the lack of precise quantitative information on the taxon-specific magnitude and spatial heterogeneity of these diel vertical migrations. A low [...]."*

We agree with Reviewer#2 that DVM is widespread in the zooplankton and can have significant impacts on carbon fluxes. However, the effect of active transport on the biological carbon pump is still poorly quantified at the scale of all migrating communities (Boyd et al., 2019). Quantitative data are even less available for a FFGM-specific parameterization of the DVM process and its associated fluxes of organic carbon.

The following paragraph has been added at the end of the discussion, just before the conclusion (L616):
*"Diel vertical migration (DVM) is a key process that is currently not included in the model and that could deepen the production of carcasses and fecal pellets. Recent modeling studies that accounted for DVM at the community level demonstrated significant impact of this process on carbon export (Aumont et al. 2018, Gorgues et al. 2019, Boyd et al. 2019). As some FFGM species undergo DVM (Pascual et al. 2017, Henschke et al. 2021b), this process is likely to strengthen their impact on carbon export by increasing the average depth at which carcasses and fecal pellets would be released into the water column, inducing a shorter path to the seafloor associated with lower total remineralization of these particles.»*

The corresponding section name has been changed from *"4.3.2. Carcasses and fecal pellets transfer efficiency"* to *"4.3.2. Carcasses and fecal pellets"*.

New references :

Pascual, M., Acuña, J., Sabatés, A., Raya, V., and Fuentes, V.: Contrasting diel vertical migration patterns in Salpa fusiformis populations, Journal of Plankton Research, 39, 836–842, 2017.

Boyd, P. W., Claustre, H., Levy, M., Siegel, D. A., and Weber, T.: Multi-faceted particle pumps drive carbon sequestration in the ocean, Nature, 568, 327–335, 2019.

Gorgues, T., Aumont, O., and Memery, L.: Simulated changes in the particulate carbon export efficiency due to diel vertical migration of zooplankton in the North Atlantic, Geophysical Research Letters, 46, 5387–5395, 2019.

**2. Generally the size of doliolids residing in the epi-pelagic layer is small except for one species. Therefore I consider that role of doliolids in their estimation was negligible. Is it possible to estimate the impact depending on the taxonomic group?**

The data do show a much lower median concentration for doliolids (0.01 mg C m$^{-3}$) than for salps (0.55 mg C m$^{-3}$) and pyrosomes (0.59 mg C m$^{-3}$) (taking the median of non-null biomass samples for each Order). This suggests that the role of doliolids on the carbon cycle would be negligible within large pelagic tunicates. For instance, assuming that carcasses and fecal pellets are identical within pelagic tunicates, and that the median of an Order divided by the sum of the medians of the Orders (Salpida + Pyrosomatida + Doliolida) is a good proxy of the contribution of the order to the biomass of large pelagic tunicates, doliolids would contribute to less than 1% of the production of carcasses and fecal pellets from FFGMs and thus to the associated carbon

fluxes. These are very uncertain values based on very rough assumptions, but it would be difficult to assess the impact by taxonomic group using the PISCES-FFGM model, given the lack of constraints on the parameterization of the different groups, which led to the choice to represent only one group of "Large pelagic tunicates".

**3. Please add references to the CNP ratio of pelagic tunicates in line 103.**

Following the C:N:P ratio implemented for the other zooplankton groups modelled in PISCES-v2 (Aumont et al. 2015), and to avoid variable stoichiometry (which requires a substantial increase in computational resources), the C:N:P ratio of the two present macrozooplankton groups was set to the Redfield ratio (122:16:1). This has been clarified in the manuscript as follows:

L103: *"As with micro- and mesozooplankton in the standard version of PISCES, the C:N:P stoichiometric composition of the two macrozooplankton groups is assumed to be constant."* has been replaced by *"As with micro- and mesozooplankton in the standard version of PISCES, the C:N:P stoichiometric composition of the two macrozooplankton groups is assumed to be constant and equal to the Redfield ratio (Aumont et al. 2015)."*